# IκB kinase phosphorylates cytoplasmic TDP-43 and promotes its proteasome degradation

Yohei Iguchi[1], Yuhei Takahashi[1], Jiayi Li[1], Kunihiko Araki[1,2], Yoshinobu Amakusa[1], Yu Kawakami[1], Kenta Kobayashi[4], Satoshi Yokoi[1], and Masahisa Katsuno[1,3]

Cytoplasmic aggregation of TDP-43 in neurons is a pathological feature common to amyotrophic lateral sclerosis (ALS) and frontotemporal lobar degeneration (FTLD). We demonstrate that the IκB kinase (IKK) complex promotes the degradation of cytoplasmic TDP-43 through proteasomes. While IKKβ is a major factor in TDP-43 degradation, IKKα acts as a cofactor, and NEMO functions as a scaffold for the recruitment of TDP-43 to the IKK complex. Furthermore, we identified IKKβ-induced phosphorylation sites of TDP-43 and found that phosphorylation at Thr8 and Ser92 is important for the reduction of TDP-43 by IKK. TDP-43 phosphorylation at Ser92 was detected in a pattern different from that of C-terminal phosphorylation in the pathological inclusion of ALS. IKKβ was also found to significantly reduce the expression level and toxicity of the disease-causing TDP-43 mutation. Finally, the favorable effect of IKKβ on TDP-43 aggregation was confirmed in the hippocampus of mice. IKK and the N-terminal phosphorylation of TDP-43 are potential therapeutic targets for ALS and FTLD.

## Introduction

Aggregation of specific proteins is a common feature of various neurodegenerative diseases, including Alzheimer's and Parkinson's disease. This has been suggested to play a central role in neuronal cell death (Soto and Pritzkow, 2018). Cytoplasmic aggregation of misfolded TAR DNA-binding protein-43 (TDP-43) occurs in 97% of cases of amyotrophic lateral sclerosis (ALS) and almost half of frontotemporal lobar degeneration (FTLD; Ling et al., 2013; Tan et al., 2017). These are collectively known as TDP-43 proteinopathies (Neumann et al., 2006; Arai et al., 2006).

TDP-43 in the pathological inclusion is hyperphosphorylated, ubiquitinated, and fragmented (Hasegawa et al., 2008). Pathological TDP-43 phosphorylation at the C-terminals is observed in the neurons and glial cells of TDP-43 proteinopathies (Neumann et al., 2009; Hasegawa et al., 2008). Phosphospecific TDP-43 antibodies against the C-terminal epitopes are standard diagnostic tools for ALS and FTLD (Hasegawa et al., 2008). However, no certain conclusions have yet been drawn regarding the effect of TDP-43 protein phosphorylation on the pathogenesis of TDP-43 proteinopathy (Liachko et al., 2010; Iguchi et al., 2012; Nonaka et al., 2016; Wu et al., 2021). Furthermore, studies on the phosphorylation of TDP-43 other than those of C-terminal remain scarce.

Nuclear factor kappa-B (NF-κB) is a transcription factor that regulates multiple aspects of innate and adaptive immune functions. Multiple extracellular signals lead to NF-κB activation, including cytokines, infectious agents, and oxidants. Most signals that trigger the NF-κB signaling pathway converge on the activation of a molecular complex containing IκB kinases (IKKs) such as IKKα, IKKβ, and NEMO. NF-κB is also speculated to play a certain role in ALS. TDP-43 was demonstrated to interact with NF-κB, and an NF-κB inhibitor reduced ALS disease symptoms in a TDP-43 transgenic mouse model (Swarup et al., 2011). In addition, the lifespan of mutant SOD1 mice was prolonged by microglia-specific inhibition of NF-κB pathway (Frakes et al., 2014). These data suggest a certain role for the NF-κB pathway in the pathogenesis of ALS. In turn, it is reported that NF-κB immunoreactivity tends to be absent from neuronal nucleus but increased in microglia (Sako et al., 2012). While these reports suggested that the NF-κB activation in microglia contributes to the disease progression via neuroinflammation, the role of the NF-κB signaling pathway in neurons is still unclear. Because TDP-43 pathological changes mainly occur in neurons, it would be necessary to analyze the effect of the NF-κB IKK signaling pathway on TDP-43 aggregation in neurons.

We herein demonstrate that a component of IKK complex, IKKβ, phosphorylates the N-terminus of TDP-43 and promotes TDP-43 degradation via the proteasome pathway. IKKβ does not affect the expression of wild-type (WT) TDP-43 but reduces that of the cytoplasmic aggregation-prone TDP-43. This favorable

[1]Department of Neurology, Nagoya University Graduate School of Medicine, Nagoya, Japan;   [2]Medical Faculty, Institute of Experimental Epileptology and Cognition Research, University of Bonn, Bonn, Germany;   [3]Department of Clinical Research Education, Nagoya University Graduate School of Medicine, Nagoya, Japan;   [4]Section of Viral Vector Development, National Institute for Physiological Sciences, Okazaki, Japan.

Correspondence to Yohei Iguchi: iguyo@med.nagoya-u.ac.jp;   Masahisa Katsuno: ka2no@med.nagoya-u.ac.jp.

phosphorylation is observed in the subset of TDP-43 inclusion in the motor neurons of sporadic ALS.

## Results

### IKKβ reduces cytoplasmic TDP-43 aggregation

To evaluate the effects of IKKs on TDP-43 metabolism, we prepared V5-tagged WT and aggregation-prone (3A2S) TDP-43 in which nuclear localization signal (NLS) and RNA recognition motif 1 (RRM1) were mutated (Fig. 1 A; Urushitani et al., 2010; Uchida et al., 2016). While TDP-43 WT was expressed in cell nuclei, the 3A2S mutation that aggregated in cellular cytoplasm was immunopositive for ubiquitin and p62 (Fig. S1, A and B), reproducing the pathological characteristics of the motor neurons of patients with sporadic ALS. Sequential fractionation detected TDP-43 WT in all of the fractions, whereas 3A2S mutation was mainly observed in sarkosyl-soluble (Sar) and sarkosyl-insoluble (ppt) fractions (Fig. S1 C), indicating that the 3A2S mutation is an aggregation-prone TDP-43. Neuro2a cells were cotransfected with V5-TDP-43 and each Flag-tagged IKK: IKKα, IKKβ, or NEMO. In immunofluorescence staining, all IKKs are not colocalized with TDP-43 WT but well colocalized with TDP-43 3A2S because these proteins are mainly detected in the cytoplasm (Fig. 1, B and E). Immunoblotting revealed that the TDP-43 WT levels were not changed by the expression of any of the three Flag-IKKs (Fig. 1, C and D). However, the expression of IKKβ, but not IKKα or NEMO, significantly reduced the protein expression of TDP-43 3A2S (Fig. 1, F and G). No between-group difference was observed in the amount of exogenous TDP-43 in the human mRNA (Fig. 1 H). NF-κB luciferase reporter assay revealed that the overexpression of IKKβ, but not IKKα or NEMO, significantly increased NF-κB activity (Fig. 1 I). This is likely due to the different functions of each IKK. Unlike IKKα, IKKβ is indispensable for IKK activation and the induction of NF-κB DNA-binding activity in most cell types (Senftleben et al., 2001; Hu et al., 1999; McKenzie et al., 2000), whereas NEMO is a noncatalytic scaffolding protein.

### The kinase activity of IKKβ promotes the degradation of TDP-43 aggregation

Because IKKβ phosphorylates multiple substrates in addition to IκB and is involved in pathways other than NF-κB (Scheidereit, 2006), we speculated that cytoplasmic TDP-43 can be another phosphorylation target of IKKβ and that the phosphorylation possibly promotes TDP-43 degradation similar to IκB. For our evaluation of the effects of IKKβ kinase activity on TDP-43 protein metabolism, the inactive IKKβ mutation, IKKβ SA, was used as a control. NF-κB luciferase assay confirmed that IKKβ SA significantly suppresses IKKβ kinase activity (Fig. 2 A). Immunoblotting revealed that the expression of IKKβ WT significantly reduced TDP-43 3A2S levels more than the expression of mock and the IKKβ SA control (Fig. 2, B and C). The IKKβ-induced decrease in aggregation-prone TDP-43 appears to occur at the protein level because its expression in mRNA was not reduced by the IKKβ (Fig. 2 D). We evaluated endogenous TDP-43 expression under IKKβ induction and found that IKKβ did not affect the expression of endogenous TDP-43 (Fig. S2). Both the

ubiquitin–proteasome system (UPS) and autophagy are known to cause TDP-43 degradation (Urushitani et al., 2010; Scotter et al., 2014; Iguchi et al., 2016); therefore, we treated Neuro2a cells with the proteasome inhibitor, MG132, or the autophagy inhibitor, bafilomycin. Immunoblotting revealed that MG132, but not bafilomycin, increased TDP-43 3A2S in cells that were also expressing either IKKβ WT or IKK SA (Fig. 2, E–G). Cycloheximide (CHX) chase assay revealed that IKKβ WT facilitates the degradation of TDP-43 3A2S compared with IKKβ SA (Fig. 2, H and I). Furthermore, IKKβ WT significantly reduced TDP-43 mNLS mutations (Fig. 2, J and K), which are predominantly observed in the cell cytoplasm. These findings suggest that the kinase activities of IKKβ induce the proteasome degradation of cytoplasmic TDP-43. On the other hand, while the expression of IKKα WT did not change the TDP-43 3A2S levels (Fig. 1, F and G), IKKα suppression led to a significant increase in the protein expression of TDP-43 3A2S under the expression of IKKβ WT but did not under the expression of IKKβ SA (Fig. 2, L–N), suggesting that IKKα is a cofactor for the IKKβ-dependent reduction of aggregation-prone TDP-43. Next, we determined whether the activation of endogenous IKKβ can affect the protein expression of TDP-43. The cells were treated with TNFα for 6 h and the luciferase reporter assay revealed that TNFα activated NF-κB in a dose-dependent manner (Fig. 2 O). Immunoblots exhibited that TNFα did not change the expression of TDP-43 WT, whereas it reduced TDP-43 3A2S in a dose-dependent manner (Fig. 2, P–R).

### NEMO plays a critical role in the degradation of cytoplasmic TDP-43

We applied the IKKβ/NEMO complex inhibitor Shikonin (Yu et al., 2022) and found that this molecule dose-dependently increases the expression of aggregation-prone TDP-43 in the cells expressing IKKβ WT but did not in the cells with IKKβ SA (Fig. 3, A–C). The knockdown of NEMO also resulted in a significant increase in the protein expression of TDP-43 3A2S under the expression of either IKKβ WT or SA (Fig. 3, D–F), suggesting that NEMO plays an important role in the process of TDP-43 degradation. Next, we prepared IKKβ delta NEMO-binding domain (dNBD), which lacks an NBD (Fig. 3 G). The luciferase assay revealed that the deletion of NBD in IKKβ did not affect NF-κB activity (Fig. 3 H). However, experiments in which TDP-43 3A2S was coexpressed with each IKKβ showed that IKKβ dNBD failed to reduce the expression of aggregation-prone TDP-43 (Fig. 3, I and J), suggesting that NEMO is vital to the promotion of TDP-43 degradation by IKKβ. Because the zinc finger (ZF) domain in NEMO is essential to the interaction with IκBα, which is a major target of IKK (Schröfelbauer et al., 2012), we investigated the interactions between TDP-43 and NEMO with or without the ZF domain (Fig. 3 K). As a result, NEMO did not bind to TDP-43 WT (Fig. 3 L) but interacted with TDP-43 mNLS (Fig. 3, M and N). However, NEMO lacking a ZF domain (NEMO dZF) bound little to TDP-43 mNLS (Fig. 3, M and N). These data suggest that NEMO functions as a scaffold for the recruitment of cytoplasmic TDP-43 to IKKβ. TDP-43 3A2S was not evaluated as it was found to be resistant to lysis in IP lysis buffer owing to its tendency to aggregate (data not shown).

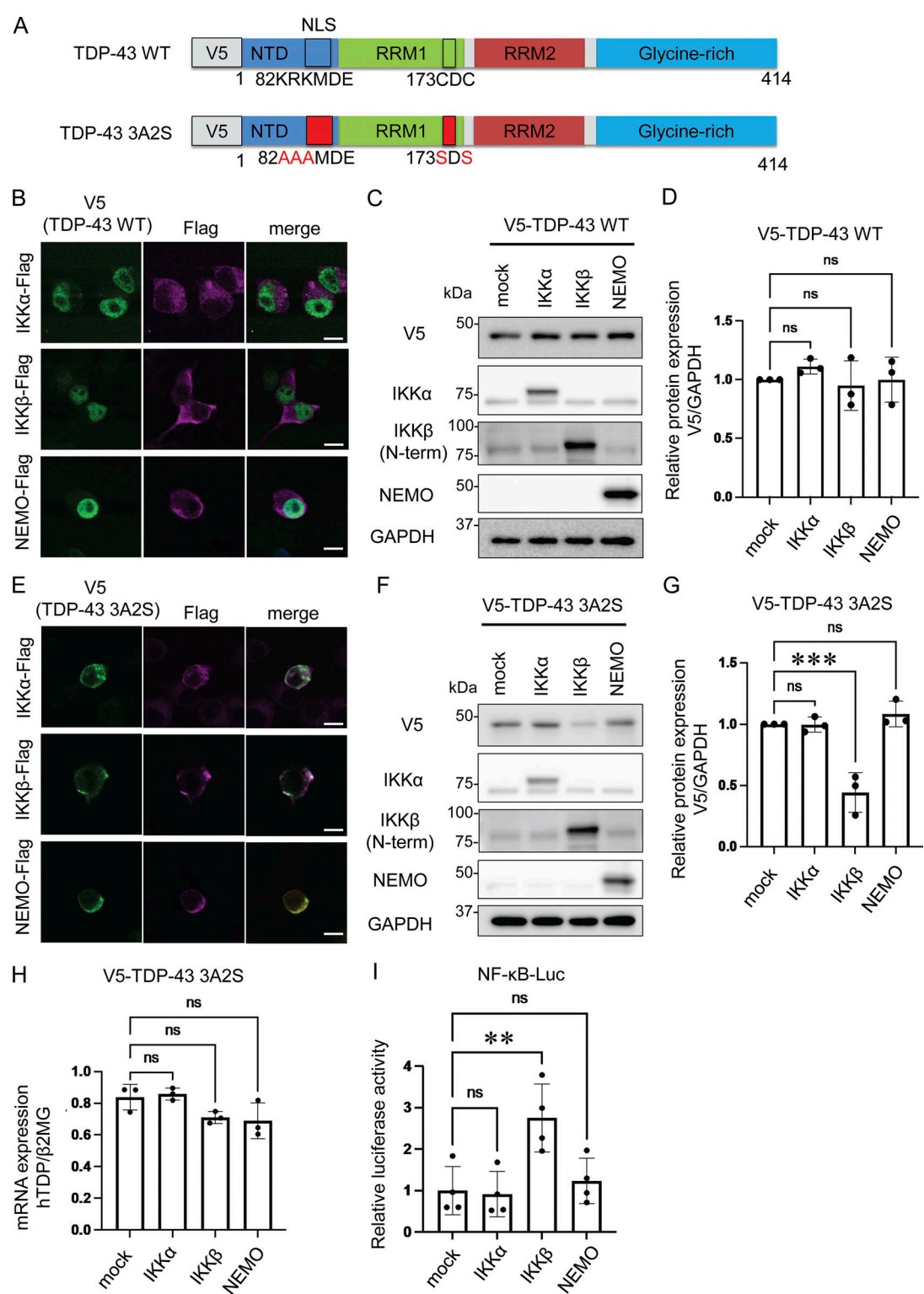

Figure 1.   **Reduction of aggregation-prone TDP-43 by IKKβ. (A)** Schematic illustrations of TDP-43 WT and the aggregation-prone TDP-43 mutation, TDP-43 3A2S, in which NLS and RRM1 are modified, respectively. **(B)** Immunofluorescence images of Neuro2a cells expressing V5-TDP-43 WT and each Flag-IKK: IKKα, IKKβ, or NEMO (green, V5; violet, Flag). Scale bars = 10 μm. **(C)** Representative immunoblots of Neuro2a cell lysates expressing V5-TDP-43 WT and each Flag-IKK. **(D and G)** Densitometric quantifications of V5 normalized with GAPDH (n = 3 for each group). **(E)** Immunofluorescence images of Neuro2a cells expressing V5-TDP-43 3A2S and each Flag-IKK (green, V5; violet, Flag). Scale bars = 10 μm. **(F)** Representative immunoblots of whole Neuro2a cell lysates expressing V5-TDP-43 3A2S and each Flag-IKK. **(H)** mRNA expression levels of human TDP-43 as measured via qRT-PCR. Data are shown as TDP-43 level-to-β2MG level ratios in human mRNA (n = 3 for each group). **(I)** Relative NF-κB activity of Neuro2a cells expressing each Flag-IKK (n = 3 for each group). In D and G–I, data were analyzed via analysis of variance (ANOVA) and Tukey's test. Error bars indicate SDs. *P < 0.05, **P < 0.01, ***P < 0.001. ANOVA, analysis of variance; β2MG, beta-2-microglobulin; CTD, C-terminal domain; GAPDH, glyceraldehyde 3-phosphate dehydrogenase; IKK, inhibitory kappa-β kinase. Source data are available for this figure: SourceData F1.

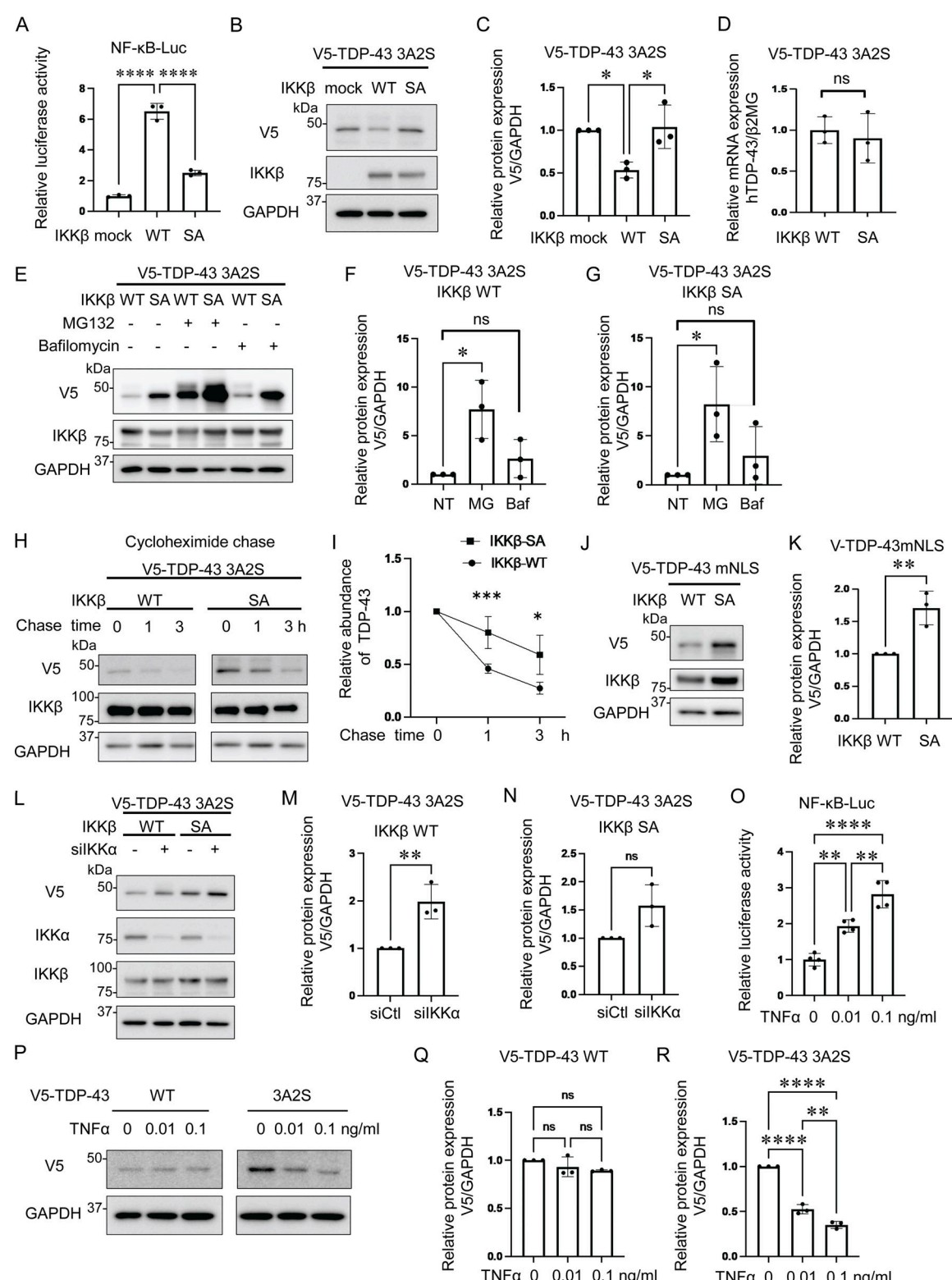

**Figure 2. Promotion of cytoplasmic TDP-43 protein degradation by IKKβ via proteasomes. (A)** Relative NF-κB activity of Neuro2a cells that express IKKβ WT or an inactive IKKβ mutation (IKKβ SA; *n* = 3 for each group). **(B)** Representative immunoblots of the lysate of Neuro2a cells expressing V5-TDP-43 3A2S and IKKβ WT or IKKβ SA. **(C)** Densitometric quantifications of V5 normalized with GAPDH (*n* = 3 for each group). **(D)** Expression levels of TDP-43 normalized with β2MG in human mRNA (*n* = 3 for each group). **(E)** Representative immunoblots of whole cell lysates from Neuro2a cells expressing V5-TDP-43 3A2S and IKKβ WT or IKKβ SA. The cells were treated with 1-mM MG132 or 100-nM bafilomycin A1 for 24 h. **(F and G)** Densitometric quantifications of V5 normalized with GAPDH (*n* = 3 for each group). **(H)** Protein degradation assay results for V5-TDP-43 3A2S with IKKβ WT or IKKβ SA. The cells were treated with 10-μM CHX. **(I)** The densitometric quantifications of V5 (*n* = 3 for each group) exhibited a relative abundance of TDP-43 3A2S. **(J)** Immunoblots of whole cell lysates from Neuro2a cells expressing V5-TDP-43 mNLS and IKKβ. **(K)** Densitometric quantifications of V5 normalized with GAPDH (*n* = 3 for each group).

**(L)** Immunoblots of whole cell lysates from Neuro2a cells expressing V5-TDP-43 3A2S, IKKβ, and siIKKα or control siRNA. **(M and N)** Densitometric quantifications of V5 normalized with GAPDH ($n$ = 3 for each group). **(O)** Relative NF-κB activity of Neuro2a cells treated with TNFα ($n$ = 3 for each group). **(P)** Immunoblots of whole cell lysates from Neuro2a cells expressing V5-TDP-43 WT or 3A2S with TNFα treatments. **(Q and R)** Densitometric quantifications of V5 normalized with GAPDH ($n$ = 3 for each group). In A, C, F, G, O, Q, and R, data were analyzed via ANOVA and Tukey's test; in D, K, M, and N, unpaired two-sided $t$ test was used; in I, two-way ANOVA was used. Error bars indicate SDs. *$P < 0.05$, **$P < 0.01$, ***$P < 0.001$, ****$P < 0.0001$, ns = not significant. Source data are available for this figure: SourceData F2.

### IKKβ phosphorylates multiple TDP-43 sites and promotes TDP-43 degradation

As we have demonstrated, IKKβ appears to phosphorylate TDP-43 and promote its degradation. This is similar to the effects of IKKs on IκB metabolism (Viatour et al., 2005). Thus, we evaluated TDP-43 phosphorylation using anti-pSer409/410 antibody, which is the most common phosphospecific antibody against TDP-43 and found that this antibody reacted with both endogenous and exogenous TDP-43 regardless of the expression of either IKKβ WT or SA (Fig. 4, A and B). In addition, immunocytochemistry revealed that anti-pSer409/410 antibody detected the cells expressing either IKKβ WT or SA without staining differences (Fig. 4 C). These data suggest that IKKβ does not specifically phosphorylate at Ser409/410. To explore the phosphorylation targets of IKKβ in TDP-43, liquid chromatography–tandem mass spectrometry (LC-MS/MS) was performed. IP was done using an anti-V5 antibody before the LC-MS/MS for enrichment (Fig. 4 A). We identified three candidate IKKβ-dependent phosphorylation sites in TDP-43, namely, Thr8, Ser92, and Ser180 (Fig. 4 D). Although 2-phosphopeptide with pSer180 and pSer183 was specific to the cells expressing IKKβ WT, Ser183 phosphorylation was detected in cells with both IKKβ WT and IKKβ SA inductions. All raw data of the proteomic analysis are available in Data S1. Furthermore, immunoblotting of the IP sample revealed polyubiquitination of TDP-43 by IKKβ (Fig. 4 A). To evaluate the effects of each TDP-43 phosphorylation, these sites were substituted with aspartic acid (Fig. 5 A). When the phosphomimetic mutations were expressed in Neuro2a cells, expression of the S92D mutation was dramatically lower than that of TDP-43 WT (Fig. 5 B). The mRNA expression of TDP-43 S92D was significantly higher than that of TDP-43 WT (Fig. 5 C), suggesting that the remarkable decrease in the cellular expression of the S92D protein resulted in compensatory expression in mRNA. Next, cells expressing TDP-43 WT or S92D were treated with MG132 or bafilomycin. Immunoblotting revealed that MG132, but not bafilomycin, increased the expressions of both TDP-43 WT and S92D (Fig. 5, D and E). In addition, the CHX chase assay showed that S92D degrades significantly faster than TDP-43 WT (Fig. 5 G). These data indicate that Ser92 phosphorylation promotes TDP-43 degradation through the proteasomes. While Ser92 is located within the NLS, the S92D mutation, like TDP-43 WT, was most often observed in cell nuclei (Fig. 5 H). Also, cell fractionation demonstrated no difference between the nucleus–cytoplasm ratios of S92D and TDP-43 WT, suggesting that Ser92 phosphorylation promotes TDP-43 degradation without disrupting NLS function (Fig. 5, I and J). On the other hand, analyses of each phosphorylation-resistant mutation of 3A2S TDP-43 revealed that the 3A2S-T8A and 3A2S-S92A mutations are resistant to IKKβ-induced

degradation, whereas the protein levels of 3A2S-S180/183A are decreased by IKKβ (Fig. 5, K–M). According to the results from the phosphomimetic and phosphorylation-resistant mutations, phosphorylations at Ser92 and, in part, at Thr8 play an important role in TDP-43 degradation induced by IKKβ.

### Novel phosphorylation-specific TDP-43 antibody reacts with cytoplasmic TDP-43 aggregation in sporadic ALS

To confirm whether IKKβ phosphorylates TDP-43, we generated a phosphorylation state-specific TDP-43 antibody against Ser92. In immunocytochemistry, the anti-pSer92 antibody detected the cytoplasm of HEK293T cells overexpressing IKKβ WT (Fig. 6 A). However, the test was negative for the cells with IKKβ SA, IKKα, and NEMO (Fig. 6, B, D, and E). Some cells with IKKβ dNBD were slightly positive for pSer92 (Fig. 6 C). Shikonin-treated cells expressing IKKβ WT were stained but faint against this antibody (Fig. 6 F). Furthermore, this antibody detected TDP-43 3A2S mutation in cells expressing IKKβ WT but did not in cells expressing IKKβ SA (Fig. 6, G and H). Immunoblotting revealed that anti-pSer92 antibody reacts within the 43 kD protein band. They are significantly increased by the overexpression of IKKβ and decreased by the knockdown of TDP-43 (Fig. 7, A–D). Immunoblotting of the cells treated with TNFα revealed the phosphorylation at Ser92 in a dose-dependent manner, whereas the expression of endogenous TDP-43 was not changed (Fig. 7, F–H). On the other hand, the suppression of IKKβ significantly reduced the level of Ser92 phosphorylation, together with the increase in endogenous TDP-43 (Fig. 7, I–K). In vitro kinase assay with recombinant protein of IKKβ and TDP-43 (1–260) revealed that IKKβ directly phosphorylated TDP-43 at Ser92 (Fig. 7 L). While anti-pSer92 antibody did not react with the endogenous nuclear TDP-43 of cultured cells or human postmortem spinal motor neurons (Fig. 6, A–F; and Fig. 8, A and B), we found that 40% of skein-like inclusions, 75% of dot-like inclusions, and 85% of round inclusions in the sporadic ALS spinal motor neurons were positive for this antibody (Fig. 8, C–E). Anti-pSer92 antibody staining mainly occurs at the center of the large inclusions, whereas anti-pSer409/410 antibody predominantly occurs at the outer edge of these inclusions (Fig. 8, C, E, and E'). This antibody never reacts with granular inclusions (Fig. 8 F), but often reacted with glial cell inclusion (80%; Fig. 8 G).

### IKKβ reduces disease-causing mutations of TDP-43

Mutations in *TARDBP* have been found to cause both sporadic and familial ALS with or without FTLD, and most are observed at glycine-rich TDP-43 C-terminals (Zou et al., 2017). The pathological features of ALS with *TARDBP* mutations are indistinguishable from those of sporadic ALS, suggesting that these

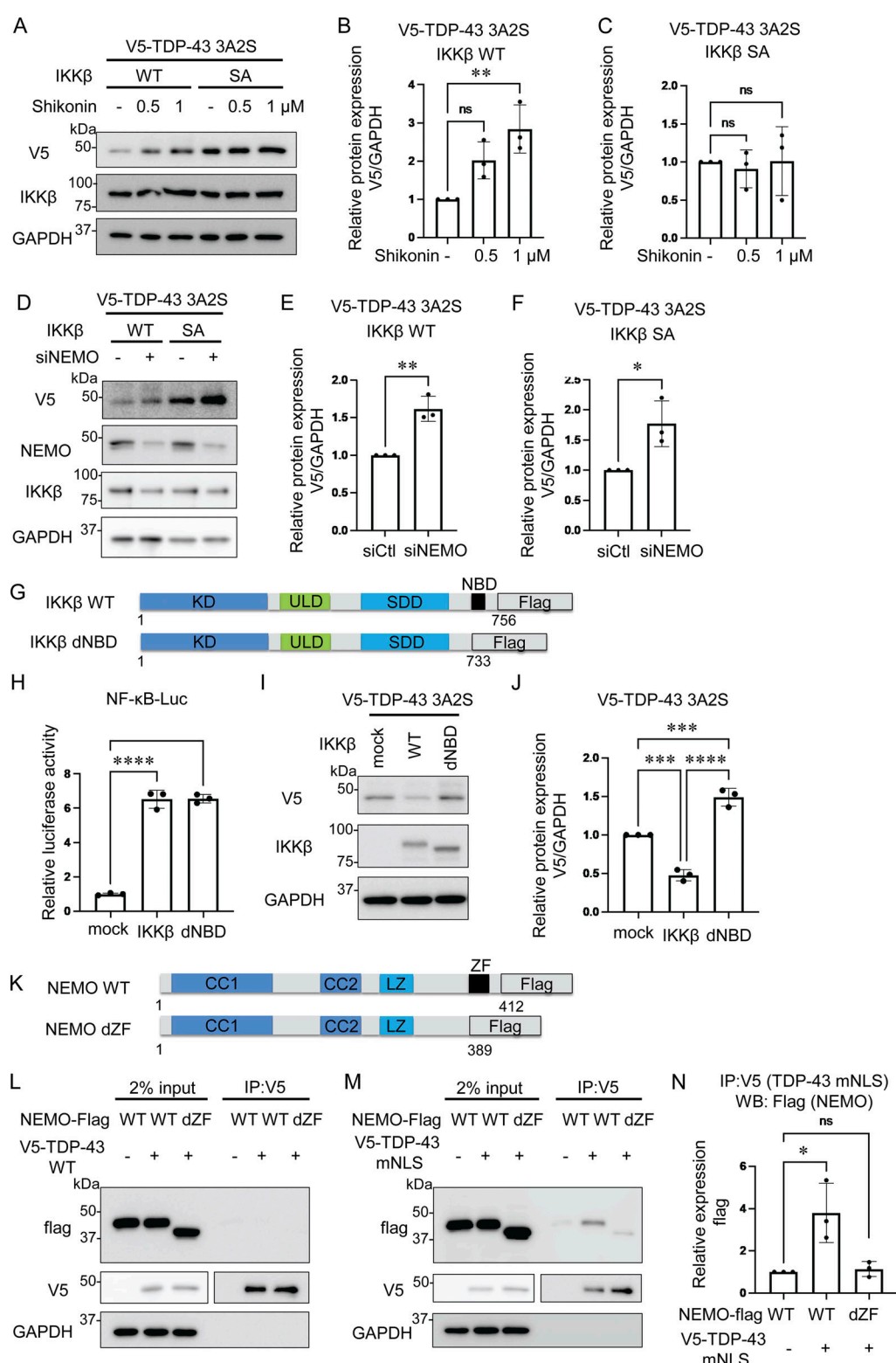

Figure 3. **Role of NEMO in the degradation of cytoplasmic TDP-43. (A)** Representative immunoblots of whole cell lysates from Neuro2a cells expressing V5-TDP-43 3A2S and IKKβ WT or IKKβ SA treated with the indicated doses of Shikonin. **(B and C)** Densitometric quantifications of V5 normalized with GAPDH ($n$ = 3 for each group). **(D)** Immunoblots of whole cell lysates from Neuro2a cells expressing V5-TDP-43 3A2S, IKKβ, and siNEMO or control siRNA. **(E and F)** Densitometric quantifications of V5 normalized with GAPDH ($n$ = 3 for each group). **(G)** Schematic illustrations of IKKβ WT, dNBD, and dCTD. **(H)** Relative NF-κB activity of Neuro2a cells expressing IKKβ WT, dNBD, or dCTD ($n$ = 3 for each group). **(I)** Representative immunoblots of whole cell lysates from Neuro2a cells expressing V5-TDP-43 3A2S and each Flag-IKK: IKKβ WT, and dNBD. **(J)** Densitometric quantifications of V5 normalized with GAPDH ($n$ = 3 for each group). **(K)** Schematic illustrations of NEMO WT and dZF. **(L and M)** Immunoblots of 2% input and immunoprecipitated proteins of Neuro2a cells expressing V5-TDP-43 WT (L)/mNLS (M) and each Flag-NEMO: WT or dZF. **(N)** Densitometric quantifications of Flags immunoprecipitated with V5-TDP-43 mNLS ($n$ = 3

for each group). In B, C, H, J, and N, data were analyzed via ANOVA and Tukey's test; in E and F, unpaired two-sided *t* test was used. Error bars indicate SDs. *P < 0.05, ****P < 0.0001, ns = not significant. CC1, coiled-coil 1; CC2, coiled-coil 2; LZ, leucine zipper; SDD, scaffold/dimerization domain; ULD, ubiquitin-like domain. Source data are available for this figure: SourceData F3.

mutations are precursors to pathological aggregation (Van Deerlin et al., 2008; Yokoseki et al., 2008). We produced six constructs of the disease-causing TDP-43 mutations at the C-terminal: A287S, A315T, A321V, M337V, G348C, and A382T (Fig. S3 A). We expressed these TDP-43 mutations in Neuro2a cells and investigated the solubilities of these mutations using 1% Triton X-100 buffer. We found that the relative ratio of the insoluble fraction of A321V was the largest in these TDP-43 mutations (Fig. S3, B and C). This is consistent with previous research demonstrating that the A321V variant of TDP-43 disrupts helix interactions, resulting in insoluble aggregates (Conicella et al., 2020). As another ALS-causing mutation, the presence of K181E adjacent to the RRM domain disrupts RNA capacity to bind to TDP-43 and enhances TDP-43 aggregation (Chen et al., 2019). Thus, we evaluated the solubility and subcellular localization of the A321V mutation with or without K181E (Fig. 9 A). While the solubility of the A321V mutation was slightly decreased, that of the A321V and K181E double mutation was significantly lower than that of TDP-43 WT (Fig. 9, B and C). In immunocytochemistry, about 20% of the cells expressing A321V or the double mutation exhibited cytoplasmic expression, whereas most of the cells expressing TDP-43 WT exhibited nuclear distributions (Fig. 9, D and E). While cytoplasmic A321V was diffusely distributed, the cytoplasmic double mutation frequently aggregated (Fig. 9 D). These disease-causing mutations were used to evaluate the effects of IKKβ. IKKβ phosphorylated

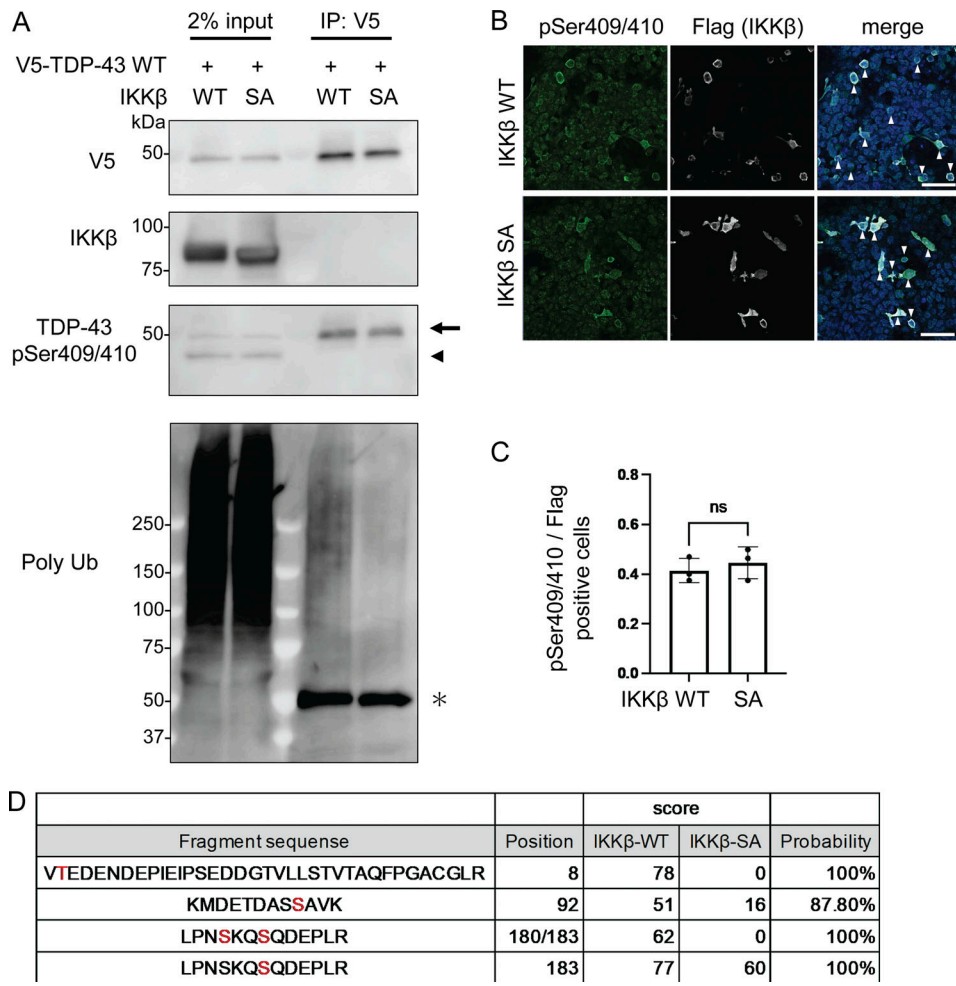

Figure 4. **IKKβ phosphorylation of multiple TDP-43 sites. (A)** Immunoblots of 2% input immunoprecipitated proteins of HEK293T cells expressing V5-TDP-43 WT and IKKβ WT or IKKβ SA. The arrowhead indicates the band endogenous TDP-43 and the arrow indicates the band of exogenous TDP-43. The asterisk indicates heavy chain IgG. **(B)** Immunofluorescence images of HEK293T cells expressing IKKβ WT or IKKβ SA (green, pSer409/410; silver, Flag; blue, DAPI). The arrowheads indicate the cells double-positive against Flag and pSer409/410. Scale bars = 50 μm. **(C)** Ratio of pSe409/410-positive cells per Flag-positive cells (*n* = 3 for each group). Unpaired two-sided *t* test was used. Error bars indicate SDs. ns = not significant. **(D)** List of the serine/threonine TDP-43 residues phosphorylated in Neuro2a cells expressing IKKβ WT or SA. These residues were identified via LC-MS/MS analysis. Source data are available for this figure: SourceData F4.

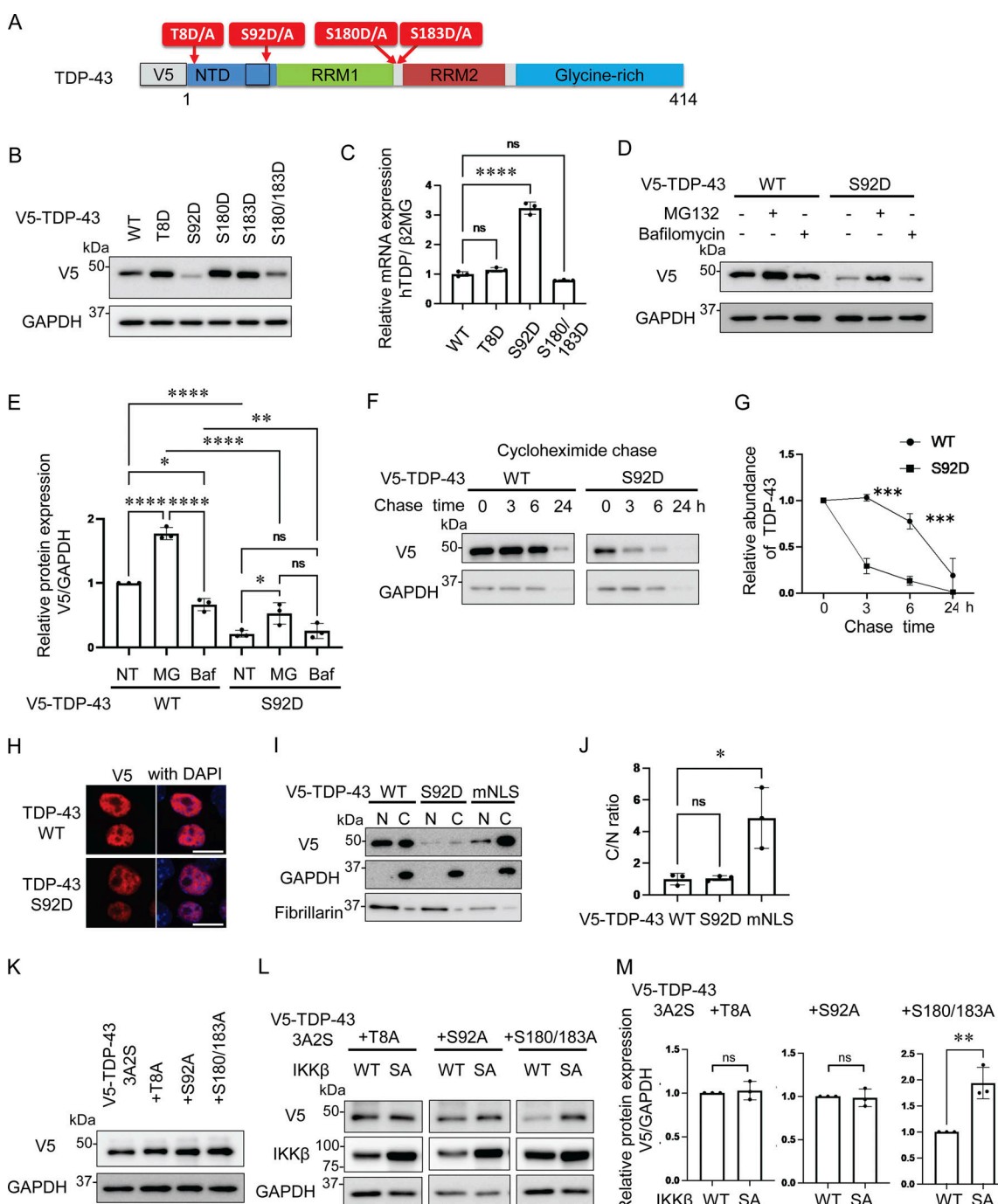

Figure 5. **Promotion of TDP-43 degradation through TDP-43 N-terminal phosphorylation. (A)** Schematic illustration of TDP-43 showing phosphorylation sites subjected to substitutions. **(B)** Immunoblots of whole cell lysates from Neuro2a cells expressing V5-TDP-43 WT or each phosphomimetic mutation. **(C)** The expression levels of TDP-43 normalized with β2MG (n = 3 for each group) in human mRNA. **(D)** Immunoblots of whole cell lysates from Neuro2a cells expressing V5-TDP-43 WT or the S92D mutation. The cells were treated with 1-mM MG132 or 100-nM bafilomycin A1 for 24 h. **(E)** Densitometric quantifications of V5 normalized with GAPDH (n = 3 for each group). **(F)** Protein degradation assays for V5-TDP-43 WT and S92D. The cells were treated with 10-μM CHX. **(G)** Densitometric quantifications of V5 (n = 3 for each group) exhibited relative abundances of TDP-43 WT and S92D, respectively. **(H)** Immunofluorescence images of Neuro2a cells expressing V5-TDP-43 WT or S92D. **(I)** Immunoblots of nuclear and cytoplasmic fractions from Neuro2a cells expressing V5-TDP-43 WT or S92D. GAPDH and fibrillarin were the cytoplasm and nucleus markers for each. **(J)** The nucleus/cytoplasm ratios were obtained through densitometric quantifications of V5 (n = 3 for each group). **(K)** Immunoblots of whole cell lysates from Neuro2a cells expressing V5-TDP-43 3A2S with each phosphorylation-resistant mutation. **(L)** Immunoblots of whole cell lysates from Neuro2a cells expressing V5-TDP-43 3A2S with each phosphorylation-resistant mutation and IKKβ WT or IKKβ SA. **(M)** Densitometric quantifications of V5 normalized with GAPDH (n = 3 for each group). In C, E, and J, data were analyzed via ANOVA and Tukey's test; in G, two-way ANOVA was used; in M, unpaired two-sided t test was used. Error bars indicate SDs. *P < 0.05, **P < 0.01, ***P < 0.001, ****P < 0.0001. Source data are available for this figure: SourceData F5.

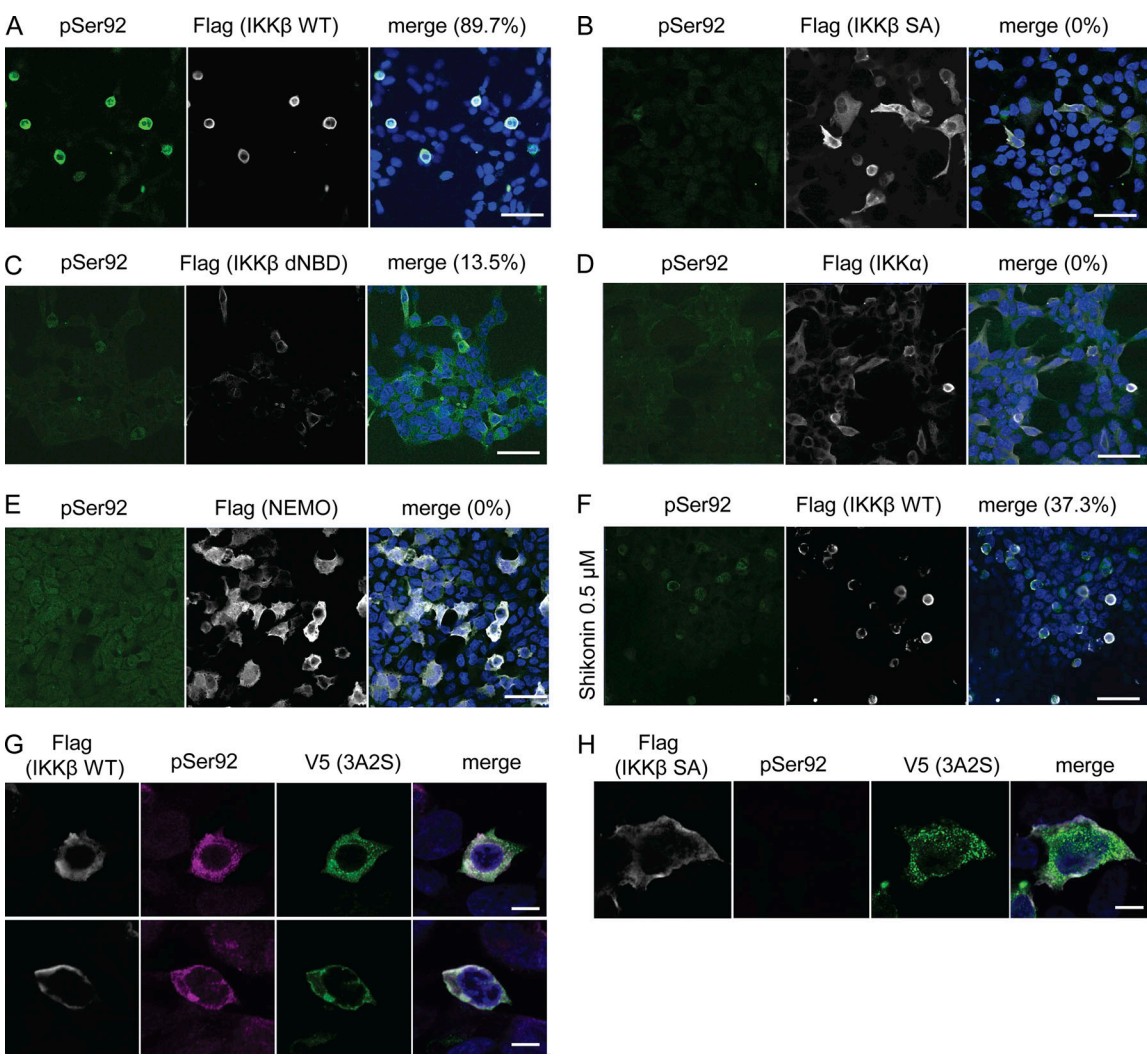

Figure 6.  **Reactions between anti-TDP-43 pS92 antibody and IKKβ-expressing cells. (A–F)** Immunofluorescence images of HEK293T cells expressing Flag-IKKβ WT (A and F), Flag-IKKβ SA (B), Flag-IKKβ dNBD (C), Flag-IKKα (D), or Flag-NEMO (E) (green, pS92; silver, Flag; blue, DAPI). Scale bars = 50 μm. **(F)** Cells expressing Flag-IKKβ WT were treated with 0.5-μM Shikonin. The percentage of PS92-positive cells in Flag-positive cells is shown. More than 100 Flag-positive cells are evaluated. **(G and H)** Immunofluorescence images of HEK293T cells expressing V5-TDP-43 3A2S and Flag-IKKβ WT or SA (green, V5; violet, pS92; silver, Flag; blue, DAPI). Scale bars = 10 μm.

TDP-43 mutations, which were expressed in the cytoplasm, but did not phosphorylate TDP-43 WT (Fig. S4). We found that IKKβ WT slightly reduces the protein expression of A321V and significantly reduces the double mutation expression (Fig. 9, F and G). Furthermore, the cell viability assay revealed that IKKβ mitigated the cell toxicity induced by the disease-causing mutation (Fig. 9 H). These data indicate that IKKβ reduces the expression and the toxicity of disease-causing mutation that reproduces the TDP-43 aggregation pathology in ALS.

## IKKβ mitigates toxicity of aggregation-prone TDP-43 in mouse brain

To evaluate the treatment efficacy of IKKβ, we conducted an in vivo experiment with neuron-specific TDP-43 knockout mouse (TDP-43flox/flox::CamKII-Cre/+: TDP-43cKO) by crossing TDP-43flox/flox mouse (Iguchi et al., 2013) and CamKII-Cre mouse, in which TDP-43 is specifically knocked out in the

neurons of the forebrain. We used TDP-43cKO mice for this experiment to mimic the pathology of TDP-43 proteinopathy. Our previous study demonstrated that TDP-43 was knocked out in the hippocampal neurons, particularly in CA1 and the dentate gyrus of the TDP-43cKO mouse (Araki et al., 2019). We inserted TDP-43 3A2S-GFP, IKKβ WT/SA-Flag sequence into the pAAV-FLEX vector (Fig. 10 A), which allowed us to express specific molecules under the condition of Cre expression. Immunohistochemistry confirmed that the exogenous TDP-43 and IKKβ were expressed in the cytoplasm of the hippocampal neurons 3 wk after the stereotaxic injection into the hippocampus of TDP-43cKO mice (Fig. 10 B). The cytoplasm of the neurons expressing IKKβ WT, but not IKKβ SA, was positive for TDP-43 pSer92 (Fig. 10 C). In the evaluation of neuronal damage, there were significantly fewer cleaved caspase 3-positive neurons in the CA1 and dentate gyrus of TDP-43cKO mice expressing IKKβ WT compared with the mice expressing IKKβ SA (Fig. 10, D and E),

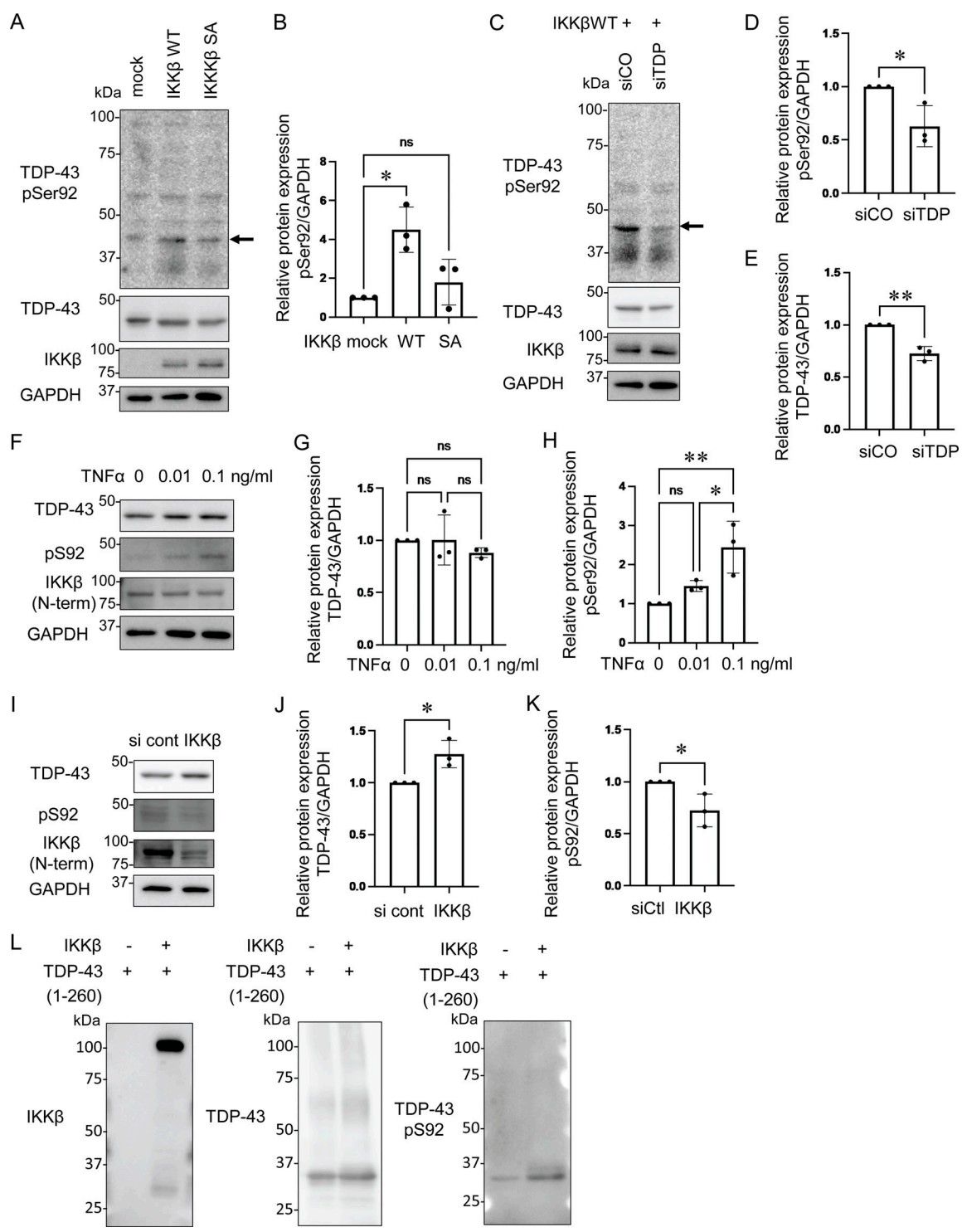

Figure 7. **TDP-43 S92 phosphorylation induced by IKKβ. (A)** Immunoblots of whole cell lysates from HEK293T cells expressing IKKβ WT or IKKβ SA. The arrow indicates TDP-43 monomer band at 43 kD. **(B)** Densitometric quantifications of pS92 normalized with GAPDH ($n$ = 3 for each group). **(C)** Immunoblots of whole cell lysates from HEK293T cells expressing IKKβ WT with control or TDP-43 siRNA. The arrow indicates the TDP-43 monomer band at 43 kD. **(D)** Densitometric quantifications of pS92 normalized with GAPDH ($n$ = 3 for each group). **(E)** Densitometric quantifications of TDP-43 normalized with GAPDH ($n$ = 3 for each group). **(F)** Immunoblots of whole cell lysates from HEK293T cells treated with TNFα. **(G)** Densitometric quantifications of TDP-43 normalized with GAPDH ($n$ = 3 for each group). **(H)** Densitometric quantifications of pS92 normalized with GAPDH ($n$ = 3 for each group). **(I)** Immunoblots of whole cell lysates from HEK293T cells treated with siIKKβ or control siRNA. **(J)** Densitometric quantifications of TDP-43 normalized with GAPDH ($n$ = 3 for each group). **(K)** Densitometric quantifications of pS92 normalized with GAPDH ($n$ = 3 for each group). **(L)** In vitro kinase assay using recombinant protein of TDP-43 (1–260) and IKKβ. In B, G, and H, data were analyzed via ANOVA and Tukey's test; in D, E, J, and K, unpaired two-sided $t$ test was used. Error bars indicate SDs. *P < 0.05, **P < 0.01. Source data are available for this figure: SourceData F7.

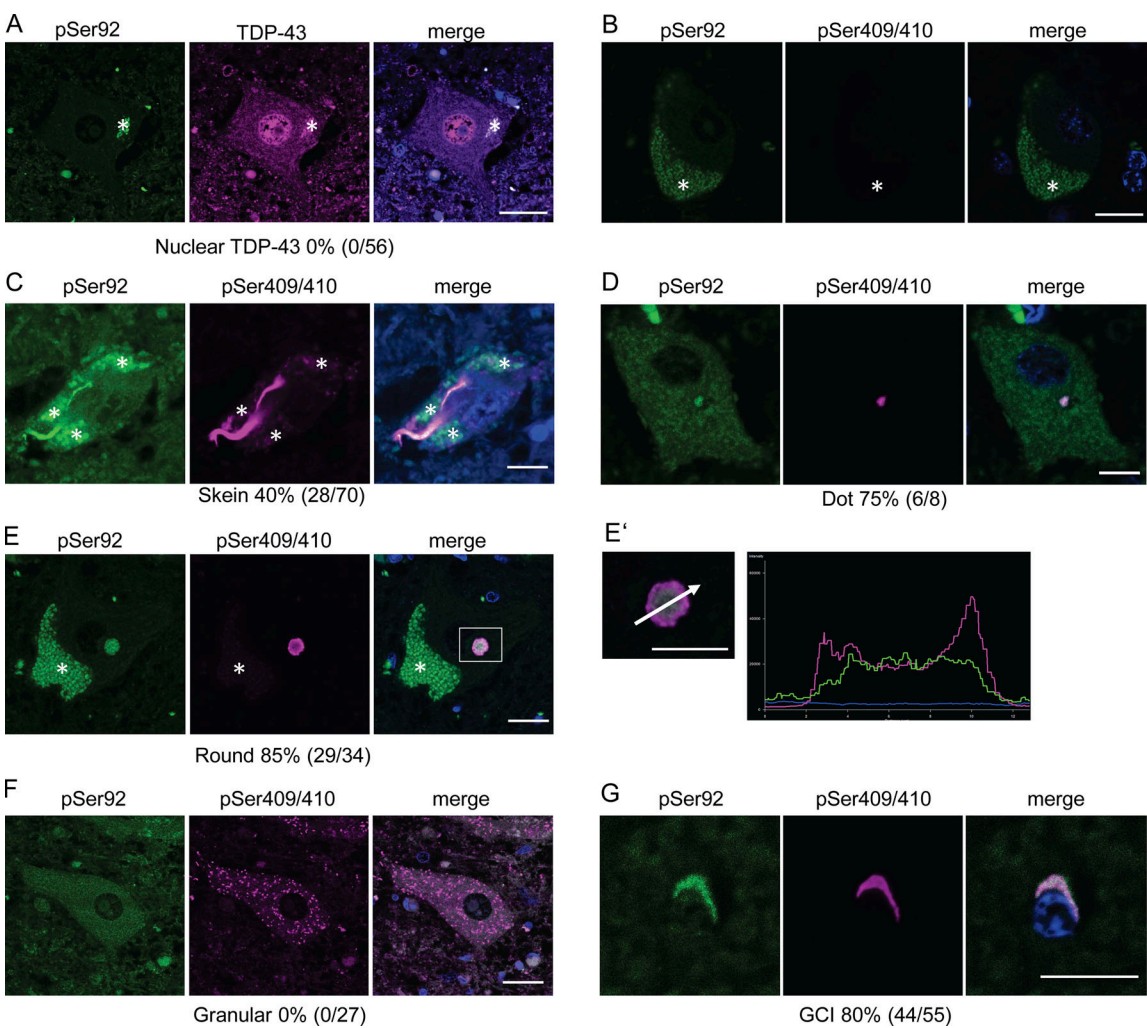

Figure 8. **Reactions between anti-TDP-43 pS92 antibody and aggregated cytoplasmic TDP-43 in sporadic ALS. (A–G)** Representative immunofluorescence images of the spinal motor neurons of controls (A and B) and patients with ALS (C–F), oligodendrocytes of patients with ALS (G) (green, pS92; violet, pS409/410; blue, DAPI). A total of 146 spinal motor neurons and 55 oligodendrocytes from five patients with sporadic ALS were analyzed. Scale bars = 10 μm. Enlarged image of the white-framed area in Fig. 8 D and a line scan analysis of the inclusion. Asterisks indicate autofluorescence of lipofuscin.

suggesting that IKKβ can mitigate the toxicity of TDP-43 cytoplasmic aggregation in neurons in vivo.

## Discussion

IKKs form a protein complex comprising kinases IKKα and IKKβ and the noncatalytic scaffolding protein NEMO. While the primary function of IKKs is the phosphorylation of IκBs to activate NF-κB, several other kinase substrates have been identified (Hinz and Scheidereit, 2014). Therefore, IKK activity is not restricted to NF-κB-dependent pathways but contributes to various other biological functions, including immune responses, transcriptional regulation, and chromatin remodeling (Scheidereit, 2006; Hinz and Scheidereit, 2014). IKKβ is critical for the activation of the NF-κB signaling pathway by inflammatory stimuli (Schmid and Birbach, 2008). Furthermore, IKKβ has been reported to phosphorylate Huntingtin (Htt) at Ser13 and enhance its clearance by proteasomes and lysosomes (Thompson et al., 2009). IKKβ knockout is known to worsen Huntington's disease

phenotype in mice, suggesting that it protects against the progression of the disease (Ochaba et al., 2019). The present study demonstrated that overexpression of IKKβ, but not IKKα or NEMO, promotes the degradation of cytoplasmic aggregation-prone TDP-43 by proteasomes. Yet, the knockdown of either IKKα or NEMO and the inhibition of the IKKβ/NEMO complex significantly disrupted the IKKβ-dependent reduction of aggregation-prone TDP-43. In addition, IKKβ that lacked NBD did not phosphorylate TDP-43 or reduce its expression despite NF-κB activation. Taking these findings together, the IKK complex actually acts on TDP-43 protein metabolism. Interestingly, the reduction of TDP-43 protein expression by IKKβ occurred specifically in 3A2S, mNLS, and the disease-causing mutation of TDP-43, all of which are prone to cytoplasmic distribution and/or aggregation. We also found that NEMO binds to cytoplasmic TDP-43 via the ZF domain. The ability of NEMO to recruit a downstream target to the IKK complex could explain why IKKβ specifically promotes cytoplasmic TDP-43 degradation. We confirmed that the activation of endogenous IKKβ with TNFα

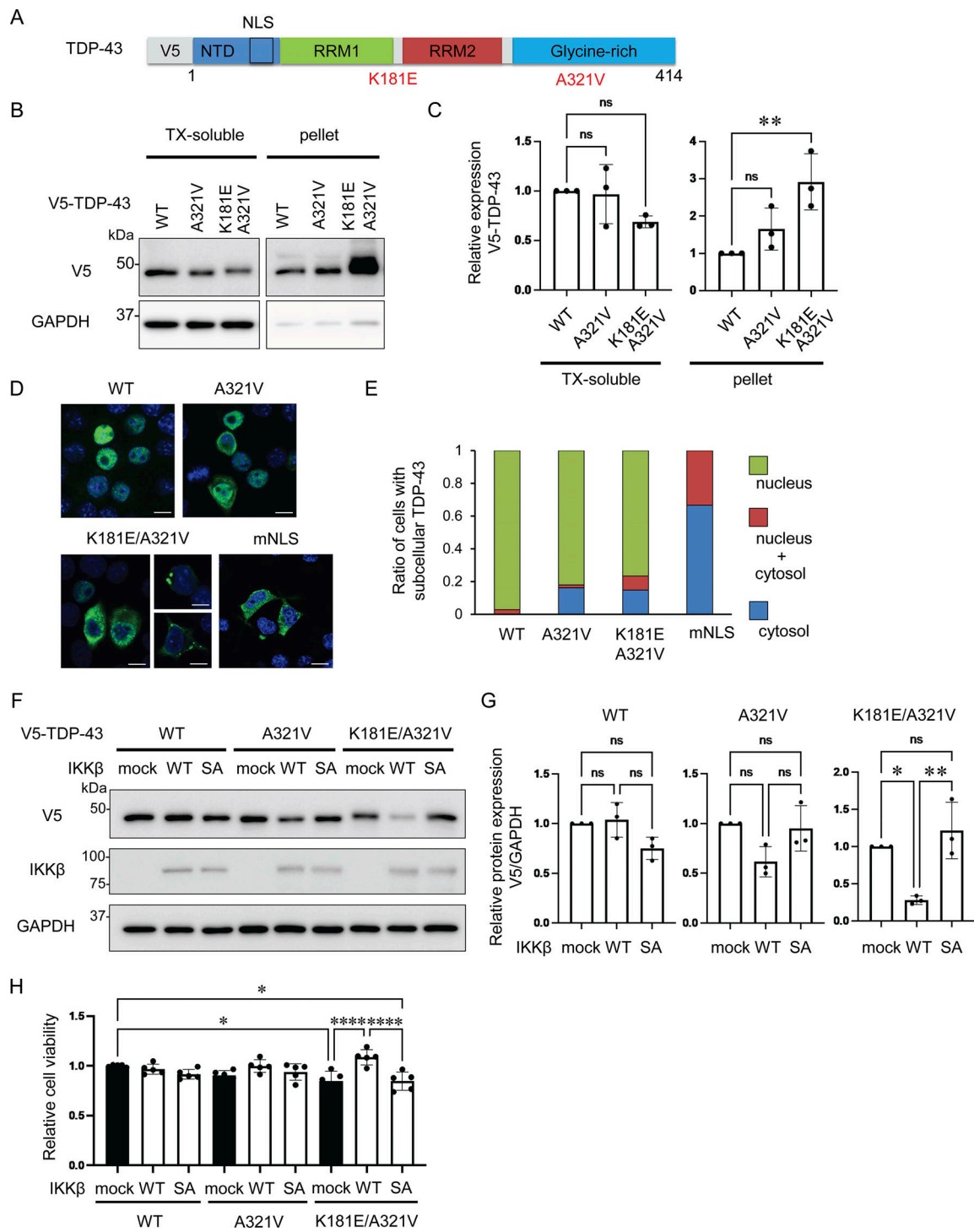

Figure 9. **IKKβ-induced reduction of disease-causing TDP-43 ALS mutations. (A)** Schematic illustration of TDP-43 with the ALS-causative missense mutations K181E and A321V. **(B)** Immunoblots of 1% Triton X-100-soluble or Triton X-100-insoluble (pellet) fractions from Neuro2a cells expressing V5-TDP-43 WT or each disease mutation. **(C)** Densitometric quantifications of V5 normalized with GAPDH (*n* = 3 for each group). **(D)** Immunofluorescence images of Neuro2a cells expressing TDP-43 WT or each disease mutation (green, V5; blue, DAPI). **(E)** Percentage of Neuro2a cells with subcellular V5-TDP-43. More than 100 cells were evaluated for each mutation. **(F)** Immunoblots of whole cell lysates from Neuro2a cells expressing each V5-TDP-43 and IKKβ WT or IKKβ SA. **(G)** Densitometric quantifications of V5 normalized with GAPDH (*n* = 3 for each group). **(H)** Viability of Neuro-2a cells quantified via MTS-based cell proliferation assays (*n* = 5 for each group). In C, G, and H, data were analyzed via ANOVA. Error bars indicate SDs. *P < 0.05, **P < 0.01, ****P < 0.0001. Source data are available for this figure: SourceData F9.

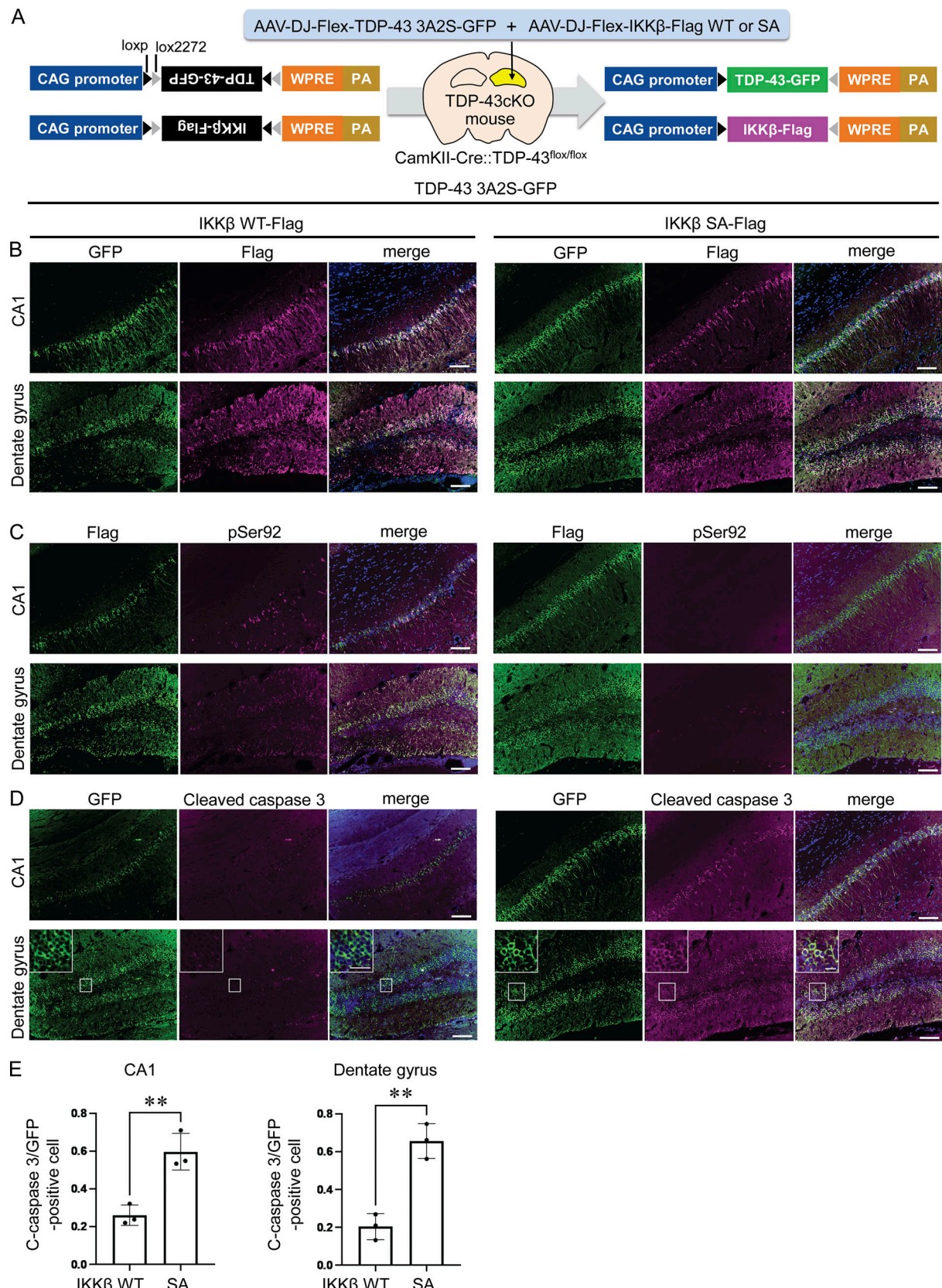

Figure 10. **Effect of IKKβ overexpression on neurons with TDP-43 pathology in vivo. (A)** Schematic illustrations of the strategy for the neuron-specific expression of TDP-43 3A2S and IKKβ WT/SA using AAV-Flex vectors and CamKII-Cre mice. TDP-43cKO (CamKII-Cre::TDP-43$^{flox/flox}$) mice were used to mimic the pathology of TDP-43 proteinopathy. **(B–D)** Immunofluorescence images of CA1 and dentate gyrus of TDP-43cKO mice (B: green, GFP; violet, Flag; blue, DAPI), (C: green, Flag; vioret, pSer92; blue, DAPI), (D: green, GFP; violet, cleaved caspase 3; blue, DAPI). Scale bars = 100 µm. The enlarged image of the framed

area is also shown. Scale bars = 20 μm. **(E)** Ratio of cleaved caspase 3-positive cells per GFP-positive cells in CA1 or dentate gyrus (*n* = 3 for each group). Unpaired two-sided *t* test was used. Error bars indicate SDs. **P < 0.01.

reduced the cytoplasmic aggregation-prone TDP-43 and phosphorylated TDP-43 at Ser92. Thus, based on these findings and the result of the in vitro kinase assay, IKKβ directly phosphorylates TDP-43 at Ser92. Although either IKKβ overexpression or activation of endogenous IKKβ does not change the expression of endogenous TDP-43, IKKβ knockdown increased endogenous TDP-43, confirming that IKKβ plays a pivotal role in the metabolism of TDP-43.

LC-MS/MS revealed that IKKβ phosphorylates TDP-43 at Thr8, Ser92, and Ser180, and our experiment, in which phosphorylation-resistant TDP-43 mutations were used, showed that phosphorylation at Thr8 and Ser92 is required in the IKKβ-dependent reduction of aggregation-prone TDP-43. We detected Ser92 phosphorylation of TDP-43 in the subset of the inclusions of ALS spinal motor neurons and observed a trend in the staining patterns of pSer92. Dot-like inclusions are frequently positive for pSer92 and can completely merge with pSer409/410, which is a standard diagnostic marker of pathological ALS inclusions. Furthermore, for relatively large, round, and skein-like inclusions, pSer92 staining occurs mainly at the center of the inclusions, whereas pSer409/410 staining mainly occurs at the outer edge of the inclusions. This suggests that the equilibrium between N-terminal and C-terminal phosphorylation is disrupted over the course of the disease, leading to the acceleration of TDP-43 aggregation in the advanced stage of ALS. However, anti-pSer92 antibodies did not detect any of the granular inclusions, indicating that the initial formation process of the granular inclusion must be fundamentally different from that of the dot-like, round, and skein-like inclusions.

This study clearly demonstrated that IKKβ reduced the amount of protein and the toxicity of TDP-43 disease mutation in the cellular model and that IKKβ mitigated the toxicity of aggregation-prone TDP-43 in the hippocampal neurons of TDP-43cKO mouse. ALS is a progressive neurodegenerative disease in which existing treatments have only modest effects on survival (Oskarsson et al., 2018). Gene silencing approaches against each amyloidogenic protein used in the treatment of neurodegenerative diseases have been developed for superoxide dismutase 1 (SOD1)-related ALS, Huntington's disease, tauopathies, and Parkinson's disease (Ghosh and Tabrizi, 2017). However, multiple lines of evidence exist against the treatment of TDP-43 proteinopathies with TDP-43 silencing strategies. Postnatal TDP-43 depletion in mouse motor neurons has been shown to cause age-dependent motor neuron degeneration, whereas nuclear TDP-43 depletion leads to a reduction in molecules indispensable for motor neurons regulated by TDP-43 via alternative splicing (Ma et al., 2022; Klim et al., 2019; Krus et al., 2022; Guerra San Juan et al., 2022; Iguchi et al., 2013). With this in mind, we propose IKK and N-terminal phosphorylation of TDP-43 as potential therapeutic targets in the treatment of TDP-43 proteinopathies that would allow reduction of toxicity, specifically, reducing the aggregation of TDP-43 or its prone protein in cellular cytoplasm, while preserving physiological nuclear TDP-43 expression levels.

In conclusion, we have presented novel evidence that TDP-43 N-terminals are targets of IKK phosphorylation. This phosphorylation facilitates the degradation of cytoplasmic TDP-43 via proteasomes. This may be an early defense mechanism against neurodegeneration in TDP-43 proteinopathies that is disrupted in the later stages of the disease process. The lines of evidence presented here are expected to contribute to the development of new disease-modifying therapy for TDP-43 proteinopathies.

## Materials and methods

### DNA constructs

V5-tagged human TDP-43 WT plasmid has been previously described (Iguchi et al., 2012). TDP-43 cDNA was amplified by PCR from the cDNA of the human spinal cord using the primers: TDP-43-F/R. The PCR product was cloned into the pENTR/D-TOPO vector (Invitrogen). An aggregation-prone variant of TDP-43 (TDP-43 2A3S) with modified RRM1 and NLS was generated with two pair sets of the primers, mRRM1-F/R and mNLS-F/R as previously described (Urushitani et al., 2010; Uchida et al., 2016). Phosphomimetic, phospho-resistant, and disease-causing mutations of TDP-43 were produced by mutagenesis with each pair of the primers T8D-F/R, S92D-F/R, S180D-F/R, S183D-F/R, T8A-F/R, S92A-F/R, S180A-F/R, S183A-F/R, A321V-F/R, and K181E-F/R. Flag-IKKα (OHu18439; GenScript), Flag-NEMO (OHu27518; GenScript), and Flag-IKKβ (#23298; Addgene) plasmids were purchased. Each mutation, IKKβ SA/dNBD/dCTD, IKKα SA, or NEMO dZF dCm, was mutagenized with the corresponding pair of the primers IKKβ SA-F/R, IKKβ dNBD-F/R, IKKβ dCTD-F/R, IKKα SA-F/R, and NEMO dZF-F/R.

### Primers

For cloning, TDP-43-F: 5′-CACCATGTCTGAATATATTCGGGT AAC-3′; TDP-43-R: 5′-CTACATTCCCCAGCCAGAAGACTTAGA AT-3′. For mutagenesis, mRRM-F: 5′-CTGACTCCAAACTTCCTA ATTCTAAGCAAAG-3′; mRRM-R: 5′-ACCATCGTCCATCTATCA TATGTCGCT-3′; mNLS-F: 5′-ACTATCCAAAAGATAACAAAC TACAAATGGATGAGACAGATGCTTC-3′; mNLS-R: 5′-GAAGCA TCTGTCTCATCCATTTGTAGTTTGTTATCTTTTGGATAGT-3′; T8D-F: 5′-CGAAGATGAGAACGATGAGCCCATTGA-3′; T8D-R: 5′-TCTACCCGAATATATTCAGACATGGTGAAG-3′; S92D-F: 5′-GATGCAGTGAAAGTGAAAAGAGCAGTCC-3′; S92D-R: 5′-TGA AGCATCTGTCTCATCCATTTTTCTTTT-3′; S180D-F: 5′-CAAAGC CAAGATGAGCCTTTGAGAAGCAGAA-3′; S180D-R: 5′-CTTATC ATTAGGAAGTTTGCAGTCACACCATCGTC-3′; S183D-F: 5′-CAA GACCAAGATGAGCCTTTGAGAAGCAGAA-3′; S183D-R: 5′-CTT AGAATTAGGAAGTTTGCAGTCACACCATCGTC-3′; T8A-F: 5′-CGAAGATGAGAACGATGAGCCCATTGA-3′; T8A-R: 5′-GCTACC CGAATATATTCAGACATGGTGAAG-3′; S92A-F: 5′-GCAGCAGTG AAAGTGAAAAGAGCAGTCC-3′; S92A-R: 5′-TGAAGCATCTGT CTCATCCATTTTTCTTTT-3′; mNLS S92A-R: 5′-TGAAGCATC TGTCTCATCCATTTGTAGTTT-3′; S180A-F: 5′-CAAAGCCAAGAT GAGCCTTTGAGAAGCAGAA-3′; S180A-R: 5′-CTTTGCATTAGG

AAGTTTGCAGTCACACCATCGTC-3′; S183A-F: 5′-CAAGCCCAA GATGAGCCTTTGAGAAGCAGAA-3′; S183A-R: 5′-CTTAGAATTA GGAAGTTTGCAGTCACACCATCGTC-3′; A321V-F: 5′-CATGATG GCTGCCGCCCAGGCA-3′; A312V-R: 5′-ACTGGATTAATGCTGAA CGCACCAAAGTT-3′; K181E-F: 5′-AGCAAAGCCAAGATGAGCCT TTGAGAAGCA-3′; K181E-R: 5′-CAGAATTAGGAAGTTTGCAGTC ACACCATCGTC-3′; IKKβSA-F: 5′-TGCACAGCATTCGTGGGGAC CCTGCAGTACCTGG-3′; IKKβSA-R: 5′-AAGAGCGCCCTGATCCA GCTCCTTGGCATATC-3′; IKKβdNBD-F: 5′-GACTACAAGGACGA CGACGACAAGGAAGC-3′; IKKβdNBD-R: 5′-ACTCTGGTCTTGTT CCCTCACAGTGTCC-3′; IKKβdCTD-F: 5′-GATTACAAGGATGAC GACGATAAGTGATAA-3′.

IKKβdCTD-R: 5′-GGCGGCTCGCTGTCCCTGCTGCA-3′; IKKαSA-F: 5′-TGTACAGCATTTGTGGGAACACTGCAGTATCTGG-3′; IKKαSA-R: 5′-CAGAGCTCCTTGATCAACATCTTTGGCATATC-3′; NEMO dZF-F: 5′-GATTACAAGGATGACGACGATAAGTGATAA-3′; NEMO dZF-R: 5′-GGGGGGGCTCCTCCTCTGGCTGGGCA-3′.

## Cell culture and transfection
Neuro2a and HEK293T cell lines were cultured in an incubator at 37°C with an atmosphere of 95% air/5% $CO_2$ in Dulbecco's Modified Eagle Medium supplemented with 10% fetal bovine serum. All cell lines were obtained from ATCC and tested negative for mycoplasma contamination. Transfection of the plasmids was performed using Lipofectamine 3000 Reagent (L3000-015; Thermo Fisher Scientific) according to the manufacturer's instructions. For the transfection of plasmid and siRNA, we used Lipofectamine 2000 (11668-019; Thermo Fisher Scientific). The siRNA sequences against mouse IKKα, NEMO, and human TDP-43, IKKβ were 5′-GAAATTTGGCACCTCCTTA-3′, 5′-GCGAGTTCAACAAGCUGAA-3′, and 5′-GACAGAUGCUUC AUCAGCAGUGAAA-3′, 5′-GCCTCACGTTTGGACATGGATCTT GT-3′, respectively. After transfection, the cells were incubated for 48 h. For intervention experiments, the cells were treated with 1-μM MG132 (M7449; Sigma-Aldrich), 100-nM bafilomycin A1 (B1793; Sigma-Aldrich), or Shikonin (14751; Cayman Chemical) 24 h after transfection. They were then harvested 48 h after transfection.

## Preparation of antibodies
Rabbit polyclonal antibody for pSer92 or pSer180 was produced by Cosmo Bio. Peptides, including TDP-43 amino acids phosphorylated at either S92, served as the epitopes for antibody production. The DETDAS(pS)AVKVKR peptide was chemically synthesized. For pS92, an extra cysteine residue was added at the N-terminal. The peptide was crosslinked to keyhole limpet hemocyanin, which is among the most commonly used peptide-binding carriers for antibody production. The rabbit antisera were purified using affinity columns. The specificities of the antibodies were verified via enzyme-linked immunosorbent assay.

## Neuron-specific TDP-43 knockout (TDP-43cKO) mice
We crossed Tardbpfl/fl mice (Iguchi et al., 2013) with CamkII-Cre mice (Tsien et al., 1996; gift from Tsuyoshi Miyakawa, Fujita Health University, Toyoake, Japan) and generated DP-43flox/flox::CamKII-Cre/+ mice as neuron-specific Tardbp knockout mice, as previously described (Araki et al., 2019).

## Stereotaxic injection of AAV vector
For neuron-specific expression in mice, TDP-43 3A2S-GFP and IKKβ-Flag WT/SA sequences were inserted into the pAAV-Flex vector (#28304; Addgene). AAV vectors were packaged using the AAV Helper Free Expression System (Cell Biolabs, Inc.). The packaging plasmids (pAAV-DJ and pHelper) and transfer plasmid were transfected into HEK293 T cells using the calcium phosphate method. The medium was replaced 18 h after transfection with fresh medium and cells were incubated for 48 h. Harvested cells were lysed by repeated freezing and thawing, and a crude cell extract containing AAV vector particles was obtained. AAV vector particles were purified by serial ultracentrifugation with cesium chloride as follows: 200,000 $g$ for 18 h with a Beckman VTi 50 rotor followed by 200,000 $g$ for 40 h with a Beckman SW 40Ti rotor. The purified particles were dialyzed with PBS and then concentrated by ultrafiltration using an Amicon 10K MWCO filter (Merck Millipore). The copy number of the viral genome (vg) was determined by the TaqMan Universal Master Mix II (Applied Biosystems) with the pair of primers: 5′-CCGTTGTCAGGCAACGTG-3′ and 5′-AGCTGACAG GTGGTGGCAAT-3′, and the fluorescent probe: 5′-TGCTGACGC AACCCCCACTGGT-3′. Real-time quantitative PCR was performed in duplicate samples using the StepOne real-time PCR system. The concentrations of the AAV vectors were $3.3 \times 10^{10}$ vg/μl for AAV-FLEX-TDP-43 3A2S-GFP and $1.6 \times 10^{10}$ vg/μl for AAV-FLEX-IKKβ WT and SA. Mice at 50 wk old were deeply anesthetized with medetomidine (0.3 mg/kg), midazolam (4 mg/kg), and butorphanol (5 mg/kg) intraperitoneally and immobilized in a stereotaxic frame. Using a Hamilton syringe, all mice were injected with AAV-FLEX-TDP-43 3A2S-GFP ($8.25 \times 10^9$ vg) and AAV-FLEX-IKKβ WT/SA ($4.0 \times 10^9$ vg) with a microinfusion pump (0.5 μl at 0.5 μl/min) into the right hippocampus of a TDP-43cKO mouse (anterior–posterior, –2.18 mm; medial–lateral, +2.01 mm; dorsal–ventral, –2.00 mm). 3 wk after the injection, the mice were sacrificed, and then immunohistochemical analyses were conducted.

## Immunocytochemistry
Cells were cultured on coverslips, fixed with 4% paraformaldehyde buffer for 30 min, and incubated with 1% Triton X-100 (9036-19-5; Sigma-Aldrich) and phosphate-buffered saline (PBS) for 5 min. The washed cells were then blocked with Tris-NaCl-blocking (TNB) buffer (FP1012; PerkinElmer) for 30 min and incubated overnight at 4°C with the primary antibodies. After further washing, the cells were incubated with the secondary antibody for 30 min and mounted with ProLong Gold Antifade Reagent with DAPI (P36935; Thermo Fisher Scientific). Confocal images were taken using ZEN (black edition) 2.1 (Carl Zeiss) on a confocal system (LSM880; Carl Zeiss) with an objective Plan-Apochromat 40×/1.4 oil immersion objective. Images are acquired using ZEN (black edition) 2.3 software. As secondary antibodies, the following antibodies were used: AlexaFluor goat anti-mouse IgG 488 (A11001; Thermo Fisher Scientific), AlexaFluor goat anti-rabbit IgG 488 (A11008; Thermo Fisher Scientific), AlexaFluor goat anti-mouse IgG 568 (A11031; Thermo Fisher Scientific), and AlexaFluor goat anti-rabbit IgG 555 (A27039; Thermo Fisher Scientific), and they were used at 1:1,000.

## Immunofluorescence of postmortem human tissue and mouse

Postmortem human lumbar spinal cord tissue specimens were obtained via autopsy. Anonymized clinical patient data are as follows: control 1, corticobasal degeneration, 74 y/o, female; control 2, brain hemorrhage, 75 y/o, male; control 3, Parkinson's disease, 77 y/o, male; control 4, multiple system atrophy, 75 y/o, male; ALS 1, 68 y/o, male; ALS 2, 85 y/o, male; ALS 3, 69 y/o, male; ALS 4, 68 y/o, male; ALS 5, 68 y/o, female. Written informed consent from the next-of-kin of each deceased patient was obtained before taking postmortem biological samples. This study was conducted in accordance with the Japanese government's Ethical Guidelines for Human Genome/Gene Analysis Research and Ethical Guidelines for Medical and Health Research Involving Human Subjects. The study and the human biological samples used were approved by the Ethics Review Committee of Nagoya University. The ALS diagnoses of patients from whom tissue samples were obtained were confirmed using both the Awaji criteria for ALS and the revised El Escorial criteria for ALS. They were also confirmed through histopathological confirmation of the presence of TDP-43 pathology. For mice, we perfused 20 ml of a 4% paraformaldehyde fixative in phosphate buffer through the left cardiac ventricle of anesthetized mice.

For immunofluorescence analyses, the tissues were postfixed overnight in 10% phosphate-buffered formalin and then processed for paraffin embedding. Tissue sections (3-mm thick) were deparaffinized and then dehydrated with alcohol. The sections were microwaved for 20 min in 50-mM citrate buffer (pH 6.0), treated with TNB buffer for 30 min, and incubated overnight at 4°C with the primary antibodies. The sections were incubated at room temperature with fluorescent secondary antibodies for 30 min and mounted with ProLong Gold Antifade Reagent with DAPI. Between each of these steps, the sections were washed with PBS without detergent three times. Image data were collected using the laser confocal microscope as described in "Immunocytochemistry."

## Quantitative real-time polymerase chain reaction

The TDP-43 levels in the mRNA of the samples were measured via reverse transcription-quantitative polymerase chain reaction (RT-qPCR). Total RNA was isolated from the cell pellets using RNeasy Mini Kit (74104; Qiagen) according to the manufacturer's instructions. The total RNA was reverse-transcribed using the SuperScript VILO cDNA Synthesis Kit (11755050; Thermo Fisher Scientific). Then, RT-qPCR was performed using the QuantiTect SYBR Green PCR Kit (204141; Qiagen) and the iCycler detection system (170-8740; Bio-Rad). The primer sequences are 5′-CCGCATGTCAGCCAAATACAAG-3′ and 5′-ACCAGAATTGGCTCCAACAACAG-3′ for human TDP-43, 5′-CTGACCGGCCTGTATGCTAT-3′ and 5′-CCGTTCTTCAGCATTTGGAT-3′ for mouse Beta-2 microglobulin (β2MG).

## Immunoprecipitation

Neuro2a cells were cultured in 6-cm dishes, washed with PBS, and lysed with 2-ml immunoprecipitation (IP) lysis buffer (25-mM HEPES-KOH, 150-mM NaCl, 5-mM MgCl2, 1-mM ethylenediaminetetraacetic acid [EDTA], 1% NP-40, pH 7.4, protease and phosphatase inhibitor cocktail [78440; Thermo Fisher

Scientific]). The lysates were centrifuged at 12,000 × $g$ for 10 min at 4°C. The supernatant was incubated on a rotator overnight with anti-V5-tag mAb-magnetic beads (M167-11; MBL) at 4°C. The beads were washed three times with PBS, eluted with sodium dodecyl sulfate (SDS) sample buffer at 95°C for 5 min, and then subjected to immunoblotting.

## Immunoblotting

For the analysis of whole cell lysates, Neuro2a and HEK293T cells were each cultured in separate 24-well plates and collected with 200-µl cell lysis buffer (50-mM Tris-HCl buffer, pH 7.5, 0.15 M NaCl, 5-mM EDTA, 1% sarkosyl, 3% SDS, protease and phosphatase inhibitor cocktail, and 2-mercaptoethanol), sonicated, and then boiled for 10 min. For protein solubility analysis, the cells cultured in 24-well plates were lysed in 200-µl TX buffer (50-mM Tris-HCl buffer, pH 7.5, 0.15 M NaCl, 5-mM EDTA, 1% Triton X-100, protease and phosphatase inhibitor cocktail), sonicated, and then centrifuged at 100,000 × $g$ for 15 min. The insoluble pellets were lysed in 50 µl of 3% SDS buffer containing 2-mercaptoethanol. NE-PER nuclear cytoplasmic reagents (78833; Thermo Fisher Scientific) were used to analyze the nucleus/cytoplasm ratios. Neuro2a cells were cultured in 24-well plates, collected, and then extracted to cytoplasmic and nuclear fractions with 200-µl cytoplasmic extraction reagent and 100-µl nuclear extraction reagent according to the manufacturer's instructions. Each 10 µl of lysate was loaded and separated by SDS-polyacrylamide gel electrophoresis. The proteins were then transferred to Hybond-P membranes (GE10600023; Amersham Pharmacia Biotech) and the membranes were blocked with 5% skimmed milk in Tris-buffered saline containing 0.05% Tween-20 and incubated with the primary antibodies overnight at 4°C. Then, each membrane was washed three times and subsequently complexed with appropriate horseradish peroxidase-conjugated secondary antibodies. Signals were detected using the ECL Western Blotting Detection System (GE Healthcare Life Sciences) and protein expression was quantified with a LAS-3000 Luminescent Image Analyzer (GE Healthcare Life Sciences). The protein bands were quantified using ImageJ software (Ver. 1.53a).

## Antibodies

Immunofluorescent and immunohistochemistry, V5 rabbit polyclonal (13202; 1:1,000; Cell signaling); V5 mouse monoclonal (R960-25; 1:1,000; Invitrogen); V5 chicken polyclonal (A190-118A; 1:1,000; Bethyl Laboratories); and Flag mouse monoclonal (F1804; 1:1,000; Sigma-Aldrich).

TDP-43 rabbit polyclonal (10782-2-AP; 1:1,000; Proteintech); TDP-43 pSer409/410 mouse monoclonal (TIP-PTD-M01; 1:1,000; Cosmo Bio); TDP-43 pSer409/410 rabbit polyclonal (TIP-PTD-P01; 1:1,000; Cosmo Bio); and Ubiquitin mouse monoclonal (D058-3; 1:1,000; MBL).

p62 rabbit polyclonal (PM045; 1:1,000; MBL); TDP-43 pSer92 rabbit polyclonal (our own; 1:1,000); GFP mouse monoclonal (M048-3; 1:1,000; MBL); and cleaved-caspase 3 rabbit polyclonal (9661L; 1:1,000; Cell signaling).

Immunoblotting, V5 rabbit monoclonal (13202; 1:2,000; Cell signaling); GAPDH mouse monoclonal (ab8245; 1:5,000; Abcam);

IKKα rabbit polyclonal (ab32041; 1:1,000; Abcam); IKKβ rabbit polyclonal (ab32135; 1:2,000; Abcam); IKKβ(N-term) rabbit polyclonal (15649-1-AP; 1:1,000; Proteintech); and NEMO rabbit monoclonal (ab178872; 1:1,000; Abcam).

TDP-43 pSer409/410 mouse monoclonal (TIP-PTD-M01; 1:1,000; Cosmo bio); Multi Ubiquitin mouse monoclonal (D058-3; 1:1,000; MBL); Fibrillarin rabbit polyclonal (ab166630; 1:1,000; Abcam); TDP-43 rabbit polyclonal (10782-2-AP; 1:1,000; Proteintech); TDP-43 (C-terminal) rabbit polyclonal (12892-1-AP; 1:1,000; Proteintech); and TDP-43 pSer92 rabbit polyclonal (our own; 1:1,000).

### Mass spectrometry

HEK293T cells were transfected with V5-TDP-43 WT and either Flag-IKKβ, WT, or SA mutation and then incubated for 48 h. The cell lysates in the IP lysis buffer were incubated overnight with anti-V5-tag mAb-magnetic beads at 4°C. V5-TDP-43 was immunoprecipitated according to the manufacturer's instructions. After reduction and alkylation, the proteins were digested in trypsin for 16 h at 37°C. The peptides were analyzed via LC-MS/MS using an Orbitrap Fusion mass spectrometer (Thermo Fisher Scientific) coupled to an UltiMate3000 RSLCnano LC system (Dionex Co.) and also a 150-mm × 75-μm ID nano-HPLC capillary column (Nikkyo Technos Co.) through a nanoelectrospray ion source. Reversed-phase chromatography was performed with a linear gradient (0 min, 5% B; 100 min, 40% B) of solvent A (2% acetonitrile with 0.1% formic acid) and solvent B (95% acetonitrile with 0.1% formic acid) and an estimated flow rate of 300 nl/min.

Before MS/MS analysis, a precursor ion scan was performed using a 400–1,600 mass-to-charge ratio (m/z). Tandem MS was performed by isolation width of 1.6 m/z with the quadrupole, HCD fragmentation with a normalized collision energy of 35%, and rapid scan MS analysis in the iontrap. Only those precursors with charge states of 2–6 were sampled for the MS2. The dynamic exclusion duration was set to 15 s with a 10-ppm tolerance. The instrument was run with 3-s cycles in the top speed mode.

The raw data were processed using Proteome Discoverer version 1.4 (Thermo Fisher Scientific) in conjunction with the MASCOT search engine version 2.7.0 (Matrix Science Inc.), which was utilized for protein identification. Peptides and proteins were identified using the UniProt human protein database (released March 2020), with a 10-ppm precursor mass tolerance and a 0.8-D fragment ion mass tolerance. The carbamidomethylation of cysteine was set as a fixed modification, and the oxidation of methionine and phosphorylation of serine and threonine were set as variable modifications. Two missed cleavages by trypsin were allowed.

### Pulse-chase assay

The medium of the Neuro2a cells was changed to an alternative containing 10-μg/ml CHX (C4859; Sigma-Aldrich) 24 h after the transfection. The cell lysates were then harvested with lysis buffer over the time lapse, and then Western blot analysis was conducted.

### Luciferase reporter assay

The Neuro2a cells seeded into 96-well plates were transfected with pNL3.2 NF-κB-RE NF-κB reporter (NlucP/NF-κB-RE/Hygro) vector (N1111; Promega) together with IKKα- and IKKβ- or Flag-NEMO vector. Luciferase assays were performed using the Dual-Glo Luciferase Assay System (E2920; Promega) 48 h after transfection according to the manufacturer's instructions. A pGL3-control vector was cotransfected to monitor the transfection efficiency. The calculated luciferase activities were normalized using this transfection efficiency.

### In vitro kinase assay

In vitro kinase assay was conducted as previously described (Thompson et al., 2009) using 500-ng recombinant IKKβ (14-485; Sigma-Aldrich) and 3-μg human TDP-43 (1-260; ab140718; Abcam) with 5X kinase buffer: 40-mM MOPS/NaOH, pH 7.0, and 1-mM EDTA. The kinase reactions were subjected to a temperature of 30°C with light agitation for 10 min under the following conditions: 5-μl recombinant IKKβ (0.1 μg/μl), 5-μl 5X kinase buffer, 2.5-μl 1-mM ATP (10127523001; Merck), 0.5-μl 0.5 M MgAc, and 6-μl recombinant TDP-43 (0.5 μg/μl) in 6-μl water. The reaction was stopped with the addition of 3% SDS buffer containing 2-mercaptoethanol, and the protein was boiled for 10 min, followed by immunoblotting.

### Graphical representations and statistical analysis

Unpaired two-sided $t$ tests were used for comparisons between two groups, whereas one-way analysis of variance (ANOVA) followed by post hoc Tukey's multiple comparison $t$ tests was used for comparisons between three or more groups. Data distribution was assumed to be normal but this was not formally tested. All statistical analyses were conducted on GraphPad Prism version 9 for macOS (http://www.graphpad.com). All graphs were also generated with GraphPad Prism.

### Online supplemental material

Fig. S1 presents the subcellular distributions and solubilities of V5-TDP-43 WT and 3A2S mutations. Fig. S2 presents endogenous TDP-43 metabolism under IKKβ overexpression. Fig. S3 demonstrates the solubility of each disease-causing TDP-43 mutation. Fig. S4 demonstrates the immunocytochemistry of Neuro2a cells expressing TDP-43 disease mutations and IKKβ. Data S1 presents the data of proteomic analysis.

### Data availability

All data that support the conclusions are available from the authors on request and/or also available in the supplemental material.

## Acknowledgments

We would like to express our gratitude to the Division for Medical Research Engineering, Nagoya University Graduate School of Medicine for analysis with LC-MS/MS and usage of confocal microscopy.

This work was supported by Japan Society for the Promotion of Science KAKENHI (grant numbers JP20H03589 [Y. Iguchi], JP20H00527 [M. Katsuno], JP21K19443 [M. Katsuno], and JP22K19506 [Y. Iguchi]); the Japan Agency for Medical Research and Development (grant number JP21wm0425013 [M. Katsuno]); the SERIKA Fund [Y. Iguchi]; the Japan ALS Association

(Yoshio Koide grant [Y. Iguchi]); and the Takeda Science Foundation [M. Katsuno]. Open Access funding provided by Nagoya University.

Author contributions: Conceptualization, Y. Iguchi and M. Katsuno; Data curation, Y. Iguchi, Y. Takahashi, J. Li, K. Araki, Y. Amakusa, Y. Kawakami, S. Yokoi, and M. Katsuno; Formal analysis, Y. Iguchi, Y. Takahashi, and J. Li; Funding acquisition, Y. Iguchi and M. Katsuno; Investigation, Y. Iguchi, Y. Takahashi, J. Li, K. Araki, Y. Amakusa, Y. Kawakami, S. Yokoi, and M. Katsuno; Methodology, Y. Iguchi, Y. Takahashi, J. Li, K. Araki, Y. Amakusa, Y. Kawakami, S. Yokoi, and M. Katsuno; Project administration, Y. Iguchi and M. Katsuno; Resources, Y. Iguchi, K. Araki, K. Kobayashi, and M. Katsuno; Supervision, Y. Iguchi and M. Katsuno; Writing—original draft, Y. Iguchi; Writing—review & editing, M. Katsuno.

Disclosures: The authors declare no competing interests exist.

Submitted: 19 February 2023

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

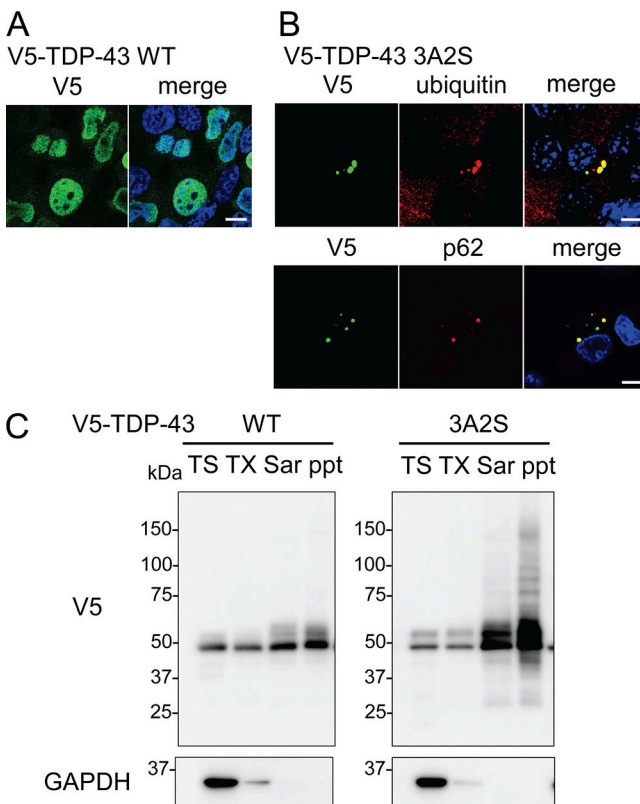

Figure S1.   **Distributions and solubilities of TDP-43 WT and 3A2S mutant. (A and B)** Immunofluorescent images of Neuro2a cells expressing V5-TDP-43 WT (A) or 3A2S (B) (green; V5, red; ubiquitin or p62, blue; DAPI). Scale bars = 10 μm. **(C)** Sequential extraction analysis of Neuro2a cells expressing V5-TDP-43 WT or 3A2S using Tris (TS), Triton X-100 (TX), Sarkosyl (Sar), and SDS buffers. Source data are available for this figure: SourceDataFS1.

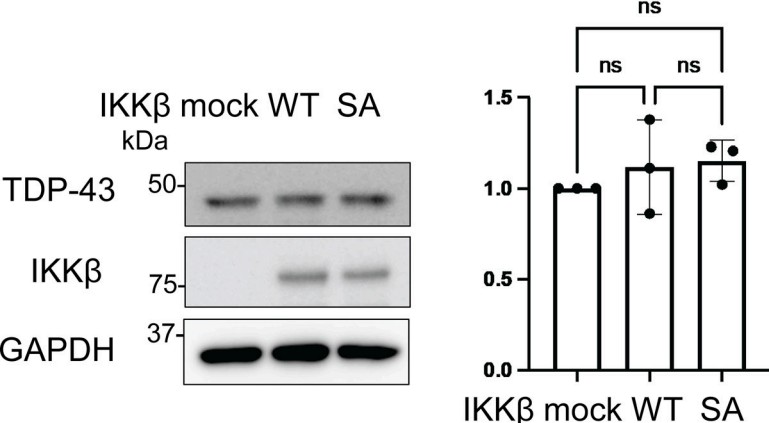

Figure S2.  **Representative immunoblots of the lysate of Neuro2a cells expressing IKKβ WT or IKKβ SA, and densitometric quantifications of TDP-43 normalized with GAPDH (*n* = 3 for each group).** Data were analyzed via ANOVA and Tukey's test. Error bars indicate SDs. Source data are available for this figure: SourceData FS2.

Figure S3. **Solubilities of TDP-43 disease-causing mutations. (A)** Schematic illustration of TDP-43 with disease-causing missense mutations. **(B)** Immunoblots of 1% Triton X-100-soluble or -insoluble (pellet) fraction from Neuro2a cells expressing V5-TDP-43 WT or each disease-mutant. **(C)** Densitometric quantifications of V5 normalized with GAPDH ($n$ = 3 for each group). Data were analyzed via ANOVA and Tukey's test. Error bars indicate SDs. *P < 0.05, ***P < 0.001. Source data are available for this figure: SourceData FS3.

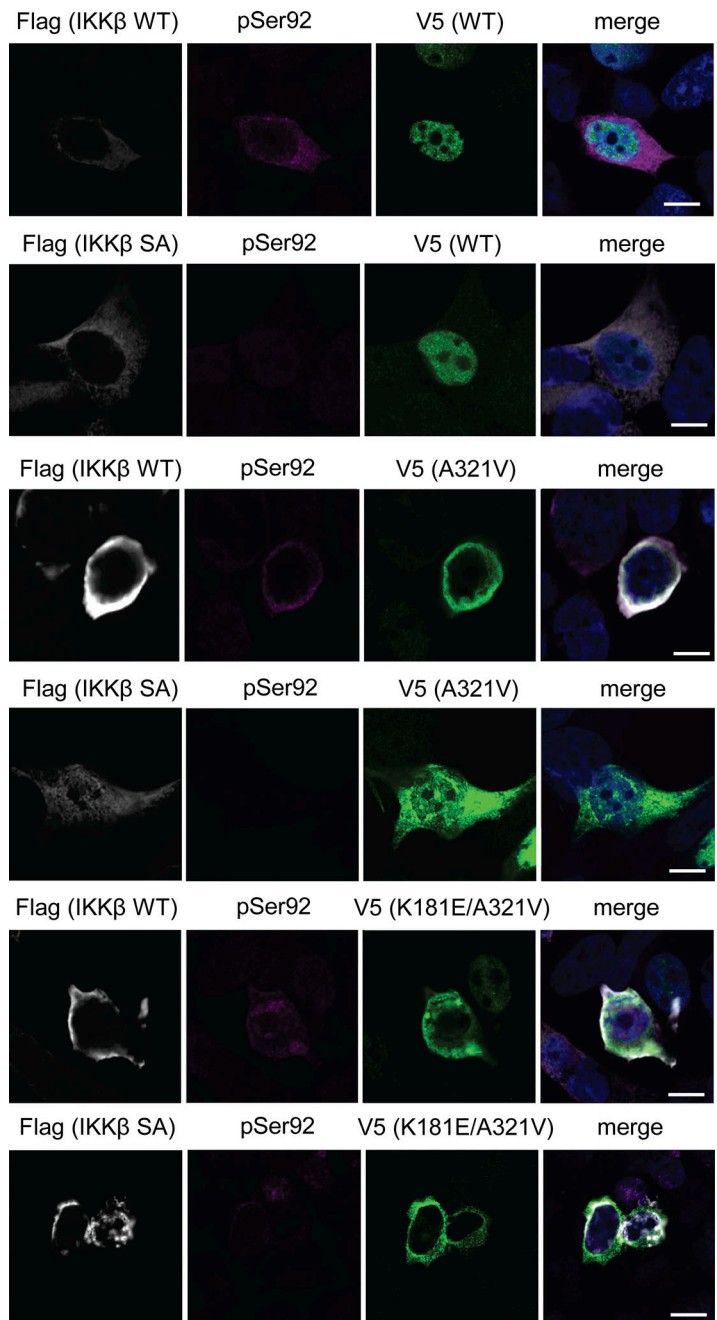

Figure S4.   **Immunofluorescent images of Neuro2a cells expressing Flag- IKKβ WT or SA and V5-TDP-43 WT or disease-mutations (A321V and K181E/A321V) (silver, Flag; violet, pSer92; green, V5; blue, DAPI).** Scale bars = 10 µm.

**Provided online is Data S1. Data S1 shows the data of proteomic analysis.**

