## [Peer Review File · The Journal of Cell Biology]

I κ B kinase phosphorylates cytoplasmic TDP-43 and promotes its proteasome degradation

Yohei Iguchi, Yuhei Takahashi, Jiayi Li, Kunihiko Araki, Yoshinobu Amakusa, Yu Kawakami, Kenta Kobayashi, Satoshi Yokoi, and Masahisa Katsuno

Corresponding Author(s): Masahisa Katsuno, Nagoya University and Yohei Iguchi, Nagoya University

Review Timeline:

Submission Date:	2023-02-19
Editorial Decision:	2023-04-05
Revision Received:	2023-10-16
Editorial Decision:	2023-11-09
Revision Received:	2023-11-17

Monitoring Editor: Zu-Hang Sheng

Scientific Editor: Tim Fessenden

Transaction Report:

DOI: <https://doi.org/10.1083/jcb.202302048>

April 5, 2023

Re: JCB manuscript #202302048

Prof. Masahisa Katsuno
Nagoya University
Neurology
65 Tsurumaicho
Showaku
Nagoya, Aichi 466-8550
Japan

Dear Prof. Katsuno,

Thank you for submitting your manuscript entitled "I κ B kinase phosphorylates cytoplasmic TDP-43 and promotes it for proteasome degradation". The manuscript has been evaluated by expert reviewers, whose reports are appended below. Unfortunately, after an assessment of the reviewer feedback, our editorial decision is against publication in JCB.

You will see that although reviewers agreed that the novel mechanism of TDP-43 modification described in this work is intriguing, they also felt that central observations made here were not appropriately validated. Resolution of these concerns would be required to confirm that the proposed interaction takes place in vivo and is relevant for disease, as laid out by Reviewer 1 (points 1-3, point 8) and confirmed by Reviewer 2. Multiple reviewers also sought greater clarity on the subcellular localization of TDP-43 and on TDP-43 knockdown studies. In addition, greater mechanistic detail was sought by Reviewer 1 on the Nemo complex. Addressing these key issues is possible but rather time-consuming and technically challenging by examining the role of endogenous IKK in neurons and validating their findings in disease models, which is beyond the regular revision timeframe. If you wish to expedite publication of the current data, it may be best to pursue publication at another journal.

Given interest in the topic, I would be open to resubmission to JCB of a significantly revised and extended manuscript that fully addresses the reviewers' concerns and is subject to further peer-review. If you would like to resubmit this work to JCB, please contact the journal office to discuss an appeal of this decision or you may submit an appeal directly through our manuscript submission system. Please include a revision plan with point by point responses to reviewer concerns with an appeal letter.

Regardless of how you choose to proceed, we hope that the comments below will prove constructive as your work progresses. We would be happy to discuss the reviewer comments further once you've had a chance to consider the points raised in this letter. You can contact the journal office with any questions, cellbio@rockefeller.edu.

Thank you for thinking of JCB as an appropriate place to publish your work.

Sincerely,

Zu-Hang Sheng
Monitoring Editor
Journal of Cell Biology

Tim Fessenden
Scientific Editor
Journal of Cell Biology

Reviewer #1 (Comments to the Authors (Required)):

In the manuscript by Iguchi et al., the authors reported that IKK β of the IKK α -IKK β -NEMO complex phosphorylated TDP-43 at S92 and S180/183, and the phosphorylation promoted the degradation of cytoplasmic aggregation-prone TDP-43 via proteasomes. This observation is interesting, suggesting that phosphorylation of TDP-43 is not always positively associated with aggregation and phosphorylation at different sites might have different regulatory effects. However, I am not fully convinced that the "novel phosphorylation" by IKK β indeed occurs endogenously since the entire study is based on overexpression of TDP-43 (WT and mutants) on Neuro2A cells, and the authors did not elaborate the physiological or pathological significance of phosphorylation of TDP-43 by IKK β at S92 and S180/183 in terms of whether it promotes or

mitigates the cytotoxicity of TDP-43. Neither did the authors provide any insight into the mechanism how N-terminal phosphorylation of TDP-43 promotes its degradation. In addition, the quality of some of the data needs to be substantially improved. As such, I am afraid that I cannot recommend the publication of this work in the Journal of Cell Biology.

Major points:

1. Page #4, line #9-10, the authors claimed that "we demonstrate that a component of I κ B kinase (IKK) complex IKK β phosphorylates the N-terminus of TDP-43 and promotes TDP-43 degradation via the proteasome pathway." However, this is merely based on the data that IKK β OE decreased the protein levels of the TDP-43-3A2S mutation. Since both the substrate (TDP-43-3A2S) and the enzyme (IKK β) are overexpressed in a cell culture system, it can very likely be an artifact. The authors need to test how endogenous TDP-43 is affected by IKK β , and how endogenous IKK β impacts WT TDP-43 phosphorylation and turnover. In other words, the authors need to: (1) knock out/down endogenous IKK β , (2) activate endogenous IKK β , and (3) determine how phosphorylation and turnover of endogenous as well as overexpressed TDP-43 (both WT and disease mutants). By the way, the authors never described in the main text what exactly the 3A2S mutation is.
2. Figure 1B, two bands are clearly seen in the IKK α western blots. The authors need to explain what they are. If the lower band is the endogenous IKK α , the lack of endogenous IKK β in IKK β 's immunoblotting indicates that IKK β is overexpressed way too much. This once again suggests the phosphorylation of TDP-43 3A2S by IKK β OE is likely an artificial effect. In addition, the author should examine the expression levels of overexpressed IKK α , IKK β and NEMO are similar and in a reasonable range compared to the endogenous proteins.
3. The authors make a very strong argument that IKK β phosphorylates TDP-43; however, the OE experiments in cell cultures can only indicate that OE of IKK β can affect the phosphorylation levels of certain overexpressed, mutant TDP-43 proteins. To claim IKK β indeed phosphorylates TDP-43, the *in vitro* kinase assay to demonstrate a direct phosphorylation of TDP-43 by IKK β is required. And the phosphorylation sites can be determined together with mass spec in this case.
4. Throughout the manuscript, the authors made quite some assumptions and speculations that are difficult to understand the rationales.
 - (1) Regarding the initiation and focus of this study, the authors never explained why they picked IKK out of hundreds of known kinases and hypothesized that IKK could phosphorylate TDP-43.
 - (2) Page 6, Line 1-3: "Since IKK phosphorylates multiple substrates in addition to I κ B and is involved in pathways other than NF- κ B (Scheidereit, 2006), we speculated that the purpose of IKK β phosphorylation of cytoplasmic TDP-43 may be to promote its degradation." I do not see how IKK phosphorylates multiple substrates can lead to the speculation that the phosphorylation of TDP-43 by IKK β would promote TDP-43 degradation. There is a huge gap in the logic.
 - (3) Page 7, Line 7-10: I do not see how an IKK β /NEMO complex inhibitor affected IKK β WT but not the enzyme-dead mutant IKK β SA could lead to the conclusion that "NEMO has an important role in the process of IKK β -induced TDP-43 degradation". To make such an argument, the authors need to knock out/down NEMO and see whether IKK β OE still affects TDP-43 phosphorylation and degradation without NEMO.
 - (4) Page 7, Line 16-18, "both IKK β dCTD and IKK β dNBD failed to reduce aggregation prone TDP-43 (Fig. 3, E and F), suggesting that NEMO is vital to the promotion of TDP-43 degradation by IKK β ." I am puzzled by the authors' experimental design. I understand the dNBD lacks the binding domain for NEMO. But, what is the purpose of including the delta C-terminal domain (dCTD) mutant (which lacks an NBD and a scaffold dimerization domain)? And, why is it important that NF- κ B activity was inhibited by IKK β dCTD but not dNBD, whereas both of them failed to reduce aggregation-prone TDP-43?
5. In Figure 5B-5D, neither T8D nor n T8A altered the protein levels of TDP-43; however, in Figure 6, 3A2S-T8A mutation was resistant to IKK β -induced degradation. How do the authors interpret these data? Does T8 phosphorylation prime that at S92? In any case, I do not think the authors can simply conclude that "the IKK β -induced TDP-43 aggregation degradation is dependent on phosphorylation at Thr8 and Ser92" based on these results (Page #9, line #18-19).
6. Figure 7 and Page #10, line #18-19, "anti-pSer92 antibody did not react with the endogenous nuclear TDP-43 of cultured cells". Can the authors explain why no pS92+ TDP-43 is detected? I ask this question is because I am not sure whether the authors' data can conclude that IKK selectively phosphorylates aggregation-prone TDP-43 or it is simply that overexpressed IKK in the cytoplasm gets the chance to phosphorylate the overexpressed TDP-43 that is abnormally located in the cytoplasm. In addition, in Figure 7A-7H, all the imaging data need to be quantified, e.g., shown as percentage of pSer92-positive cells or granules. Figure 7K, the western blots indicate the TDP-43 protein levels are not sufficiently knocked down. Again, the blots of TDP-43 should be quantified to confirm the KD efficiency of siTDP.
7. Figure 8, endogenous TDP-43 should be predominantly nuclear in the control motor neurons under normal conditions; however, the immunostaining shows strong immunoreactivity in the cytoplasm in figure 8A. And, the immunostaining of total TDP-43 should be shown. In Figure 8D, the DAPI does not register the nucleus of the motor neuron. Overall, the quality of the immunohistochemistry data is mediocre.
8. Instead of speculating that IKK β may be a modifier of TDP-43 toxicity based on that its OE affects TDP-43 phosphorylation and degradation, the authors should perform the actual experiments and determine whether and how IKK β OE, KD and inhibitor modify TDP-43 cytotoxicity. Such experiments (either in cell or animal models) are feasible and will provide not only a quick

answer but also much more insights to whether IKK is indeed involved in TDP-43 pathogenesis and possesses the potential therapeutic value.

Minor points:

1. Page #3, line #8-10: "The cytoplasmic aggregation of misfolded TAR DNA-binding protein-43 (TDP-43) occurs in 97% of cases of amyotrophic lateral sclerosis (ALS), frontotemporal lobar degeneration with TDP-43 inclusions (FTLD-TDP), primary lateral sclerosis, and progressive muscular atrophy." This is a very misleading expression. Abnormal TDP-43 protein inclusions are found in ~97% ALS but only ~45% FTD patients (Ling et al., 2013; Tan et al., 2017). The authors need to be precious and specific for each TDP-43-related disease.
2. Figure 2H and 5H, the western blots lack the loading controls.
3. Figure 3, no figure legends are provided for Figure 3I-3K.
4. Figure 9F, the blots of TDP-43 A321V show a huge difference between WT and SA, that is not registered in the quantification bar graph in Figure 9G.
5. Figure S2A-SB, the blots of total TDP-43 are not shown, which is required to evaluate the phosphorylation levels and to validate the KD efficiency of siTDP. Also, no quantification of any kind for the western blots or the image data is provided in Figure S2.

Reviewer #2 (Comments to the Authors (Required)):

The manuscript by Iguchi et al., describe the role of I κ B kinase (IKK) complex in promoting degradation of cytoplasmic aggregation-prone TDP-43 by proteasomes. The cytoplasmic mislocalization of TDP-43 in degenerating neurons is a neuropathological hallmark of amyotrophic lateral sclerosis and frontotemporal dementia. The authors found that IKK β kinase activity is important for the degradation of TDP-43 and IKK alpha acts as a cofactor for recruiting TDP-43 to the IKK complex. They went on to map the phosphorylation sites of TDP-43 and showed that Thr8 and Ser92 are necessary for the reduction of TDP-43 by IKK. Interestingly, they observed Ser92 in a different pattern from C-terminal phosphorylation in the pathological inclusion of ALS motor neurons. Importantly, they found that IKK beta drastically reduced the disease causing TDP-43 pathological aggregation in ALS. Overall, this is an interesting manuscript that provides important insights about the basic biology of TDP-43. The authors should consider addressing the following minor/moderate issues to improve the quality of their paper.

1. What about endogenous TDP-43? Most of the assays performed in the manuscript are based on overexpression system. The authors should test if the treatment with IKK modulates expression or localization of endogenous TDP-43. This is a very straightforward experiment that can be easily done.
2. While overexpressing IKK complex proteins in the N2a cells, have you examined the TDP-43 aggregation/expression by IF? The authors should perform IF to back up their Western blotting data.
3. Figure 4B lacks quantification. The authors should use arrows/circle to show the area of interest.
4. Figure 8: The authors should provide quantification data.
5. Most of the experiments are done on N2a cells only. What is the in vivo relevance of these findings? Can the authors use any existing in vivo models of TDP-43 to translate their findings to an in vivo?

Reviewer #3 (Comments to the Authors (Required)):

In this manuscript, Iguchi et al reported phosphorylation of the N-terminus of TDP-43 by IKK promotes the degradation of aggregation prone TDP-43 mutants with NEMO acting as the scaffolding protein. The levels of aggregation prone 3A2S, mNLS or A321V mutants of TDP-43 were shown to be affected by overexpression of WT but not SA mutant form of IKK-beta. In addition, they found that NEMO binds to TDP-43 and IKK-beta phosphorylates TDP-43 at T8 and S92. They generated antibodies specific to phosphorylated S92 and found accumulation of phos-S92 signals in ALS patients with TDP-43 aggregates. Overall, these are very interesting findings which could have therapeutic implications. However, the following points need to be addressed before this paper gets published.

- 1) Fig. 4 and Fig. 7; It appears that WT and endogenous TDP-43 can be efficiently phosphorylated by Ikk-beta. Why doesn't this phosphorylation lead to TDP-43 degradation? In addition, it's very strange that overexpressed 3A2S TDP-43 has very little pSer92 signals in Fig. 7G despite a huge effect of IKKbeta on its levels.
- 2) Fig. 7I: The changes in the levels of pSer92 TDP-43 is not convincing.
- 3) Fig. 7K: it's very strange that siTDP-43 has very little effect on total TDP-43 levels but has a dramatic effect on pSer92 form of TDP-43. Changes in TDP-43 total levels should be quantified too.

- 4) Does the A321V or the mNLS mutant show more phosphorylation of TDP-43 compared to WT?
- 5) Fig. 9G: the quantification does not match with the blot shown in Fig. 9F.
- 6) Fig. S3: not sure how many times the experiment has been repeated. Error bar is not shown. IT will be nice to test additional mutants besides A321V.
- 7) To determine the effect of IKKbeta on TDP-43 levels, it will be nice to include vector control transfected cells throughout the study.

Dear Dr. Sheng and Dr. Fessenden,

We are very pleased with the editorial decision of *Journal of Cell Biology* to consider a revised manuscript of our paper (202302048) entitled "*I κ B kinase phosphorylates cytoplasmic TDP-43 and promotes it for proteasome degradation*". We have addressed all issues raised by the referees and the paper has been revised accordingly.

All revised and added words are highlighted in red in the new manuscript, but we would like to highlight our major additions to the experimental data here. In response to the reviewer's suggestion, we assessed the endogenous TDP-43 metabolism under IKK β overexpression or TNF α induction (Fig.2P-R, Fig.S2, and Fig.7F), and investigated the knockdown effect of NEMO or IKK β (Fig.3D and Fig.7I). In addition, we performed *in vitro* kinase assay and confirmed that IKK β directly phosphorylates TDP-43 at Ser92. Finally, we evaluated the effect of IKK β on the toxicity induced by TDP-43 aggregation in the mouse neurons (Fig.10).

We thank you again for the opportunity to revise our manuscript and hope this revised version will now be of sufficient quality to meet the criteria for publication in the *Journal of Cell Biology*.

Yohei Iguchi, MD, PhD and Masahisa Katsuno, MD, PhD

Department of Neurology,
Nagoya University Graduate School of Medicine

New figures have been included:

Figure 1B

Figure 2B The vector control was additionally put.

Figure 2P-R

Figure 3I We put the vector control and exclude dCTD.

Figure 3D-F

Figure 3G-J IKK β dCTD was excluded.

Figure 4A Western blotting was reperformed.

Figure 4B Arrows were put.

Figure 4C

Figure 6A-F The percentage of cells positive for pSer92 was added to each picture.

Figure 6G and H The representative pictures were changed.

Figure 7D Quantification data of TDP-43 was added.

Figure 7F-L

Figure 8A and G

Figure 9F The vector control was additionally put.

Figure 9H

Figure 10A-E

Figure S2

Figure S4

Figure S5

Editor:

You will see that although reviewers agreed that the novel mechanism of TDP-43 modification described in this work is intriguing, they also felt that central observations made here were not appropriately validated. Resolution of these concerns would be required to confirm that the proposed interaction takes place in vivo and is relevant for disease, as laid out by Reviewer 1 (points 1-3, point 8) and confirmed by Reviewer 2. Multiple reviewers also sought greater clarity on the subcellular localization of TDP-43 and on TDP-43 knockdown studies. In addition, greater mechanistic detail was sought by Reviewer 1 on the Nemo complex.

Response

We performed multiple experiments including in vivo analysis as the reviewers requested. In addition, we analyzed the subcellular localization of IKK β and TDP-43 with immunofluorescent analyses shown in Figure 1, and endogenous IKK β activation (Figure 2P and 7F) and IKK β knockdown studies (Figure 7E) were also done. Furthermore, we investigated the role of NEMO on TDP-43 metabolism with NEMO knockdown as shown in Figure 3D. We are confident that this manuscript is now satisfactory to all of the reviewers.

Reviewer 1:

In the manuscript by Iguchi et al., the authors reported that IKK β of the IKK α -IKK β -NEMO complex phosphorylated TDP-43 at S92 and S180/183, and the phosphorylation promoted the degradation of cytoplasmic aggregation-prone TDP-43 via proteasomes. This observation is interesting, suggesting that phosphorylation of TDP-43 is not always positively associated with aggregation and phosphorylation at different sites might have different regulatory effects.

However, I am not fully convinced that the "novel phosphorylation" by IKK β indeed occurs endogenously since the entire study is based on overexpression of TDP-43 (WT and mutants) on Neuro2A cells, and the authors did not elaborate the physiological or pathological significance of phosphorylation of TDP-43 by IKK β at S92 and S180/183 in terms of whether it promotes or mitigates the cytotoxicity of TDP-43. Neither did the authors provide any insight into the mechanism how N-terminal phosphorylation of TDP-43 promotes its degradation. In addition, the quality of some of the data needs to be substantially improved. As such, I am afraid that I cannot recommend the publication of this work in the Journal of Cell Biology.

In response to the reviewer's criticism, we evaluated endogenous TDP-43 expression level under IKK β overexpression, IKK β knockdown, or activation of endogenous IKK β . In addition, we further investigated the effect of IKK β on cell toxicity in vivo. The details are written below in response to the reviewer's specific comments. Many data were now improved or corrected as noted by the reviewer. We hope this manuscript is appropriate for this journal now.

Major points:

1. Page #4, line #9-10, the authors claimed that "we demonstrate that a component of I κ B kinase (IKK) complex IKK β phosphorylates the N-terminus of TDP-43 and promotes TDP-43 degradation via the proteasome pathway." However, this is merely based on the data that IKK β OE decreased the protein levels of the TDP-43-3A2S mutation. Since both the substrate (TDP-43-3A2S) and the enzyme (IKK β) are overexpressed in a cell culture system, it can very likely be an artifact. The authors need to test how endogenous TDP-43 is affected by IKK β , and how endogenous IKK β impacts WT TDP-43 phosphorylation and turnover. In other words, the authors need to: (1) knock out/down endogenous IKK β , (2) activate endogenous IKK β , and (3) determine how phosphorylation and turnover of endogenous as well as overexpressed TDP-43 (both WT and disease mutants). By the way, the authors never described in the main text what exactly the 3A2S mutation is.

Response 1-1

We thank the Reviewer for raising these important points. We used TNF α to activate endogenous IKK β , and found that TNF α significantly reduced TDP-43 3A2S but not TDP-43 WT (Figure 2P). In addition, we evaluated the endogenous TDP-43 expression. Endogenous TDP-43 was not changed by the expression of IKK β (Figure S2 and Figure 7A) or by the TNF α induction (Figure 7F), although endogenous TDP-43 was phosphorylated at Ser92 by TNF α (Figure 7F). However, IKK β knockdown significantly increased endogenous TDP-43 together with the decrease of pSer92 (Figure 7I). Since endogenous TDP-43 is mainly expressed mainly in the nucleus as well as exogenous TDP-43 WT, it may be less susceptible to cytoplasmic localization of IKK β . However, IKK β depletion affects the expression of endogenous TDP-43, suggesting that IKK β has a pivotal role even in the metabolism of endogenous TDP-43. In addition, we mentioned about 3A2S mutation in the first sentence of the result part (page 5, line 17-19).

2. Figure 1B, two bands are clearly seen in the IKK α western blots. The authors need to explain what they are. If the lower band is the endogenous IKK α , the lack of endogenous

IKK β in IKK β 's immunoblotting indicates that IKK β is overexpressed way too much. This once again suggests the phosphorylation of TDP-43 3A2S by IKK β OE is likely an artificial effect. In addition, the author should examine the expression levels of overexpressed IKK α , IKK β and NEMO are similar and in a reasonable range compared to the endogenous proteins.

Response 1-2

The lower bands in the IKK α blots are endogenous IKK α , which was confirmed by the knockdown of IKK α in Figure 2L. About IKK β , we reperformed the immunoblots with a different antibody, IKK β (N-term), and that antibody recognized the endogenous bands. Now we revised Figure 1C and 1F with this antibody. We also used this antibody in Figure 7F and 7I, and confirmed the bands are endogenous IKK β by knocking-down using IKK β siRNA. In addition to this result, the effect of IKK β on TDP-43 metabolism would not be an artifact of the overexpression of IKK β , because of the following reasons.

1. The endogenous IKK β activation with TNF α decreased TDP-43 3A2S (Figure 2P-R).
2. Either IKK β SA (inactive mutant) or IKK β dNBD overexpression did not affect the expression of TDP-43 3A2S (Figure 2B and 3I).
3. Knockdown of endogenous IKK β increased endogenous TDP-43 (Figure 7I).

3. The authors make a very strong argument that IKK β phosphorylates TDP-43; however, the OE experiments in cell cultures can only indicate that OE of IKK β can affect the phosphorylation levels of certain overexpressed, mutant TDP-43 proteins. To claim IKK β indeed phosphorylates TDP-43, the in vitro kinase assay to demonstrate a direct phosphorylation of TDP-43 by IKK β is required. And the phosphorylation sites can be determined together with mass spec in this case.

Response 1-3

We would like to thank the reviewer for this important point. According to the reviewer's suggestion, an in vitro kinase assay with recombinant proteins of IKK β and TDP-43 was performed, and we now confirmed that IKK β directly phosphorylates TDP-43 at Ser92 (Figure 7L).

4. Throughout the manuscript, the authors made quite some assumptions and speculations that are difficult to understand the rationales.

(1) Regarding the initiation and focus of this study, the authors never explained why they picked IKK out of hundreds of known kinases and hypothesized that IKK could phosphorylate TDP-43.

Response 1-4-1

We now explained the reason why we focused on IKKs in terms of TDP-43 metabolism in the introduction (page 4, line 9 – page 5, line 6).

(2) Page 6, Line 1-3: "Since IKK phosphorylates multiple substrates in addition to IκB and is involved in pathways other than NF-κB (Scheidereit, 2006), we speculated that the purpose of IKKβ phosphorylation of cytoplasmic TDP-43 may be to promote its degradation." I do not see how IKK phosphorylates multiple substrates can lead to the speculation that the phosphorylation of TDP-43 by IKKβ would promote TDP-43 degradation. There is a huge gap in the logic.

Response 1-4-2

In response to the reviewer's comment, we revised this sentence as shown below (page 7, line 2-5).

"Because IKKβ phosphorylates multiple substrates in addition to IκB and is involved in pathways other than NF-κB (Scheidereit, 2006), we speculated that cytoplasmic TDP-43 can be another phosphorylation target of IKKβ, and that the phosphorylation possibly promotes TDP-43 degradation similar to IκB."

(3) Page 7, Line 7-10: I do not see how an IKKβ/NEMO complex inhibitor affected IKKβ WT but not the enzyme-dead mutant IKKβ SA could lead to the conclusion that "NEMO has an important role in the process of IKKβ-induced TDP-43 degradation". To make such an argument, the authors need to knock out/down NEMO and see whether IKKβ OE still affects TDP-43 phosphorylation and degradation without NEMO.

Response 1-4-3

We silenced NEMO and found that the knockdown of NEMO resulted in a significant increase in the protein expression of TDP-43 3A2S under the expression of IKKβ WT (Figure 3D-F). This suggests that NEMO has an important role in the TDP-43 degradation induced by IKKβ. NEMO knockdown also increased TDP-43 3A2S under the expression of IKKβ SA, suggesting that disruption of the interaction between NEMO and endogenous IKKβ affected the metabolism of cytosolic TDP-43 independent of exogenous IKKβ expression.

(4) Page 7, Line 16-18, "both IKKβ dCTD and IKKβ dNBD failed to reduce aggregation prone TDP-43 (Fig. 3, E and F), suggesting that NEMO is vital to the promotion of TDP-43

degradation by IKK β ." I am puzzled by the authors' experimental design. I understand the dNBD lacks the binding domain for NEMO. But, what is the purpose of including the delta C-terminal domain (dCTD) mutant (which lacks an NBD and a scaffold dimerization domain)? And, why is it important that NF- κ B activity was inhibited by IKK β dCTD but not dNBD, whereas both of them failed to reduce aggregation-prone TDP-43?

Response 1-4-4

NEMO is necessary for the phosphorylation of TDP-43, but IKK β can activate NF- κ B without NEMO interaction through phosphorylation of the alternative substrates, p65 and p105 (Schröfelbauer et al., 2012). As the reviewer suggested, dCTD is not necessarily needed for our conclusion. Thus, we removed the result of dCTD from Figure 3I-J.

5. In Figure 5B-5D, neither T8D nor n T8A altered the protein levels of TDP-43; however, in Figure 6, 3A2S-T8A mutation was resistant to IKK β -induced degradation. How do the authors interpret these data? Does T8 phosphorylation prime that at S92? In any case, I do not think the authors can simply conclude that "the IKK β -induced TDP-43 aggregation degradation is dependent on phosphorylation at Thr8 and Ser92" based on these results (Page #9, line #18-19).

Response 1-5

The experiment of phosphomimetic TDP-43 mutations suggests that the phosphorylation at Ser92 but not Thr8 is important for its degradation. However, both phosphoresistant mutations were resistant to the degradation of the aggregation-prone TDP-43. Phosphomimetic mutation does not necessarily exhibit a structural change similar to the phosphorylated state, whereas phosphoresistant mutations are completely dephosphorylated. These data suggest that T8D mutation may not recapitulate the phosphorylated state of TDP-43. Now we corrected the overstatement in the text (page 11, line 13-15).

6. Figure 7 and Page #10, line #18-19, "anti-pSer92 antibody did not react with the endogenous nuclear TDP-43 of cultured cells". Can the authors explain why no pS92+ TDP-43 is detected? I ask this question is because I am not sure whether the authors' data can conclude that IKK selectively phosphorylates aggregation-prone TDP-43 or it is simply that overexpressed IKK in the cytoplasm gets the chance to phosphorylate the overexpressed TDP-43 that is abnormally located in the cytoplasm.

Response 1-6-1

According to the results of this study, in physiological conditions, pSer92 level is below the detection sensitivity in immunocytochemistry. However, when IKK β is overexpressed in the cytoplasm, the phosphorylation becomes detectable. We showed that IKK β decreased cytoplasmic TDP-43, in which only NLS is mutated (Figure 2J). Thus, the effect of IKK β may not be specific to the aggregation-prone TDP-43.

In addition, in Figure 7A-7H, all the imaging data need to be quantified, e.g., shown as percentage of pSer92-positive cells or granules.

Response 1-6-2

Now, we quantified the rate of the pS92 positive cells as shown in Figure 6A-F.

Figure 7K, the western blots indicate the TDP-43 protein levels are not sufficiently knocked down. Again, the blots of TDP-43 should be quantified to confirm the KD efficiency of siTDP.

Response 1-6-3

We quantified the TDP-43 expression and found that TDP-43 was significantly reduced by the siRNA (Figure 7C and E). It may be possible that phosphorylated TDP-43 at Ser92 is more easily knocked down than non-phosphorylated TDP-43.

7. Figure 8, endogenous TDP-43 should be predominantly nuclear in the control motor neurons under normal conditions; however, the immunostaining shows strong immunoreactivity in the cytoplasm in figure 8A.

Response 1-7-1

Generally, motor neurons in elderly people contain lipofuscin in the cytoplasm which has autofluorescence. Now, the lipofuscin is marked with asterisks.

And, the immunostaining of total TDP-43 should be shown.

Response 1-7-2

Co-immunostaining with panTDP-43 and pS92 antibodies is shown in Figure 8A.

In Figure 8D, the DAPI does not register the nucleus of the motor neuron.

Response 1-7-3

We often see a weak DAPI signal in ALS motor neurons because of neurodegeneration.

Overall, the quality of the immunohistochemistry data is mediocre.

Response 1-7-4

As mentioned above, we improved the quality of immunohistochemistry. Only C-terminal phosphorylations of TDP-43 have been demonstrated in ALS pathology. This is the first report about Ser92-TDP-43 phosphorylation in the aggregation of ALS. For relatively large inclusions, pSer92 staining occurs primarily in the center of the inclusions, suggesting that the clearance system of the cytoplasmic TDP-43 may be disrupted over the course of the disease, leading to the acceleration of TDP-43 aggregation in the advanced stage of ALS.

8. Instead of speculating that IKK β may be a modifier of TDP-43 toxicity based on that its OE affects TDP-43 phosphorylation and degradation, the authors should perform the actual experiments and determine whether and how IKK β OE, KD and inhibitor modify TDP-43 cytotoxicity. Such experiments (either in cell or animal models) are feasible and will provide not only a quick answer but also much more insights to whether IKK is indeed involved in TDP-43 pathogenesis and possesses the potential therapeutic value.

Response 1-8

We evaluated the cell viability of Neuro2a cells and found that IKK β mitigated the cell toxicity induced by TDP-43 disease mutant (Figure 9H). Finally, we expressed IKK β and TDP-43 3A2S simultaneously in the hippocampus of TDP-43cKO mice and exhibited the favorable effect of IKK β (Figure 10).

Minor points:

1. Page #3, line #8-10: "The cytoplasmic aggregation of misfolded TAR DNA-binding protein-43 (TDP-43) occurs in 97% of cases of amyotrophic lateral sclerosis (ALS), frontotemporal lobar degeneration with TDP-43 inclusions (FTLD-TDP), primary lateral sclerosis, and progressive muscular atrophy." This is a very misleading expression. Abnormal TDP-43 protein inclusions are found in ~97% ALS but only ~45% FTD patients (Ling et al., 2013; Tan et al., 2017). The authors need to be precious and specific for each TDP-43-related disease.

Response

We revised the manuscript as shown in line 9-10, page 3.

2. *Figure 2H and 5H, the western blots lack the loading controls.*

Response

We put GAPDH of these immunoblots as the loading control.

3. *Figure 3, no figure legends are provided for Figure 3I-3K.*

Response

We thank the reviewer for pointing out these errors. We now added the legend of them.

4. *Figure 9F, the blots of TDP-43 A321V show a huge difference between WT and SA, that is not registered in the quantification bar graph in Figure 9G.*

Response

In response to the other reviewer's comment, we added a mock for each analysis in Figure 9 F-G. Although IKK β tended to decrease TDP-43 A321V, the inter-group difference was not significant. By contrast, IKK β significantly decreased TDP-43 mutation of K181E/A321V. Since the potency of cytoplasmic distribution is not different between A321V and K181E/A321V, the ability of IKK β to degrade TDP-43 appears to be associated with the susceptibility of TDP-43 to aggregate.

5. *Figure S2A-SB, the blots of total TDP-43 are not shown, which is required to evaluate the phosphorylation levels and to validate the KD efficiency of siTDP. Also, no quantification of any kind for the wester blots or the image data is provided in Figure S2.*

Response

In vitro kinase assay revealed that TDP-43 phosphorylation was not detected by anti-pSer180 antibody as shown below. This result indicates that IKK β does not phosphorylate TDP-43 at Ser180 directly. This result was similar for the pT8 antibody. Now the data in Figure S2 of the previous version were excluded.

Reviewer #2 (Comments to the Authors (Required)):

The manuscript by Iguchi et al., describe the role of IκB kinase (IKK) complex in promoting degradation of cytoplasmic aggregation-prone TDP-43 by proteasomes. The cytoplasmic mislocalization of TDP-43 in degenerating neurons is a neuropathological hallmark of amyotrophic lateral sclerosis and frontotemporal dementia. The authors found that IKKβ kinase activity is important for the degradation of TDP-43 and IKK alpha acts as a cofactor for recruiting TDP-43 to the IKK complex. They went on to map the phosphorylation sites of TDP-43 and showed that Thr8 and Ser92 are necessary for the reduction of TDP-43 by IKK. Interestingly, they observed Ser92 in a different pattern from C-terminal phosphorylation in the pathological inclusion of ALS motor neurons. Importantly, they found that IKK beta drastically reduced the disease causing TDP-43 pathological aggregation in ALS. Overall, this is an interesting manuscript that provides important insights about the basic biology of TDP-43. The authors should consider addressing the following minor/moderate issues to improve the quality of their paper.

1. What about endogenous TDP-43? Most of the assays performed in the manuscript are based on overexpression system. The authors should test if the treatment with IKK modulates expression or localization of endogenous TDP-43. This is a very straightforward experiment that can be easily done.

Response 2-1

We additionally evaluated endogenous TDP-43 with respect to the effect of IKKβ, and found that IKKβ overexpression (Figure S2 and Figure 7A) or endogenous IKKβ activation induced by TNFα (Figure 7F) did not affect endogenous TDP-43 as well as exogenous TDP-43 WT.

However, IKK β knockdown significantly increased endogenous TDP-43 (Figure 7I). Since endogenous TDP-43 expresses mainly in the nucleus as well as exogenous V5-TDP-43 WT, it may be less susceptible to cytoplasmic IKK β . However, IKK β depletion affects the expression of endogenous TDP-43, suggesting that IKK β has a pivotal role even in the metabolism of endogenous TDP-43.

2. While overexpressing IKK complex proteins in the N2a cells, have you examined the TDP-43 aggregation/expression by IF? The authors should perform IF to back up their Western blotting data.

Response 2-2

Now, we added the IF data in Figure 1.

3. Figure 4B lacks quantification. The authors should use arrows/circle to show the area of interest.

Response 2-3

Less than half of the cells expressing both IKK β WT and SA (inactive form) are positive for pSer409/410. We put arrows to show the cells double-positive against Flag and pSer409/410 (Figure 4B). There is no difference in the rate of pSer409/410-positive cells out of Flag-positive cells between the two groups (Figure 4B). We further performed Western blotting again with the remaining lysate and a different lot of TDP-43 pSer409/410 antibody and found that this antibody reacted with both endogenous and exogenous TDP-43. However, there is no difference in the reactivity between cells expressing IKK β WT and SA in both immunocytochemistry and Western blotting, suggesting that IKK β does not specifically phosphorylate TDP-43 at pSer409/410.

4. Figure 8: The authors should provide quantification data.

Response 2-4

We put the percentage of cells positive for pSer92 into each type of inclusion.

5. Most of the experiments are done on N2a cells only. What is the in vivo relevance of these findings? Can the authors use any existing in vivo models of TDP-43 to translate their findings to an in vivo?

Response 2-5

Although cytoplasmic TDP-43 distribution is necessary to see the effect of IKK β on TDP-43-mediated toxicity, most of the in vivo models do not have cytoplasmic distribution of TDP-43 in the neurons. Thus, we performed an experiment using conditional TDP-43 knockout mouse (CamKII-Cre::TDP-43^{flox/flox}) and AAV-FLEX vectors. This model can recapitulate the pathology seen in TDP-43 proteinopathy: nuclear depletion and cytoplasmic aggregation of TDP-43 simultaneously. With this model, we finally confirmed that IKK β induces TDP-43 phosphorylation at Ser92 and mitigates the toxicity of the aggregation-prone TDP-43 in vivo (Figure 10).

Reviewer #3 (Comments to the Authors (Required)):

In this manuscript, Iguchi et al reported phosphorylation of the N-terminus of TDP-43 by IKK promotes the degradation of aggregation prone TDP-43 mutants with NEMO acting as the scaffolding protein. The levels of aggregation prone 3A2S, mNLS or A321V mutants of TDP-43 were shown to be affected by overexpression of WT but not SA mutant form of IKK-beta. In addition, they found that NEMO binds to TDP-43 and IKK-beta phosphorylates TDP-43 at T8 and S92. They generated antibodies specific to phosphorylated S92 and found accumulation of phos-S92 signals in ALS patients with TDP-43 aggregates. Overall, these are very interesting findings which could have therapeutic implications. However, the following points need to be addressed before this paper gets published.

1) Fig. 4 and Fig. 7; It appears that WT and endogenous TDP-43 can be efficiently phosphorylated by Ikk-beta. Why doesn't this phosphorylation lead to TDP-43 degradation? In addition, it's very strange that overexpressed 3A2S TDP-43 has very little pSer92 signals in Fig. 7G despite a huge effect of IKKbeta on its levels.

Response 3-1-1

We would like to thank the reviewer for this important point. As shown in Figure 1B, all IKKs localize in the cytoplasm. Since endogenous TDP-43 and overexpressed TDP-43 WT are mainly in the nucleus, IKK β may not affect the total expression level of TDP-43. However, in the response to the other reviewer's comment, we evaluated the endogenous TDP-43 under IKK β knockdown, and found that IKK β depletion increased the expression of endogenous TDP-43, suggesting that endogenous IKK β has a pivotal role even in the metabolism of endogenous TDP-43. In addition, we repeated the experiment of the immunostaining (Figure

6G and H), and now the quality is improved from the previous data.

2) *Fig. 7I: The changes in the levels of pSer92 TDP-43 is not convincing.*

Response 3-1-2

As the reviewer pointed out, the change in the level of pSer92 is not drastic. Since in the immunocytochemistry, there is a distinct difference in the reaction against pSer92, this newly developed antibody may be less suitable for WB than for immunofluorescent applications. However, the quantification analysis revealed IKK β significantly phosphorylates TDP-43 at Ser92.

3) *Fig. 7K: it's very strange that siTDP-43 has very little effect on total TDP-43 levels but has a dramatic effect on pSer92 form of TDP-43. Changes in TDP-43 total levels should be quantified too.*

Response 3-3

We quantified the TDP-43 expression and found that TDP-43 was significantly reduced by the siRNA. TDP-43 phosphorylated at Ser92 may be more susceptible to knockdown than other TDP-43 (Figure 7C-E).

4) *Does the A321V or the mNLS mutant show more phosphorylation of TDP-43 compared to WT?*

Response 3-4

Pictures of immunocytochemistry of the cells expressing TDP-43 WT, A321V, or K181E/A321V with IKK β WT or SA are shown in Figure S4. The mutant forms are phosphorylated according to their cytoplasmic localization.

5) *Fig. 9G: the quantification does not match with the blot shown in Fig. 9F.*

Response 3-5

In response to the other reviewer's comment, we added a mock for each analysis in Figure 9 F-G. Although IKK β tended to decrease TDP-43 A321V, the inter-group difference was not significant. By contrast, IKK β significantly decreased TDP-43 mutation of K181E/A321V. Since the potency of cytoplasmic distribution is not different between A321V and K181E/A321V, the ability of IKK β to degrade TDP-43 appears to be associated with the

susceptibility of TDP-43 to aggregate.

6) *Fig. S3: not sure how many times the experiment has been repeated. Error bar is not shown. IT will be nice to test additional mutants besides A321V.*

Response 3-6

We repeated the experiment and quantified the signal intensities (Figure S3C).

7) *To determine the effect of IKKbeta on TDP-43 levels, it will be nice to include vector control transfected cells throughout the study.*

Response 3-7

In response to the reviewer's comment, we put the vector control in Figure 2B, 3I, and 9F.

November 9, 2023

RE: JCB Manuscript #202302048R-A

Prof. Masahisa Katsuno
Nagoya University
Neurology
65 Tsurumaicho
Showaku
Nagoya, Aichi 466-8550
Japan

Dear Prof. Katsuno:

Thank you for submitting your revised manuscript entitled "I κ B kinase phosphorylates cytoplasmic TDP-43 and promotes it for proteasome degradation". We would be happy to publish your paper in JCB pending following the resolution of final requests made by Reviewer 3 and final revisions necessary to meet our formatting guidelines (see details below). Please attend to points 2-4 by Reviewer 3 with changes to the text.

A. MANUSCRIPT ORGANIZATION AND FORMATTING:

Full guidelines are available on our Instructions for Authors page, <http://jcb.rupress.org/submission-guidelines#revised>. Submission of a paper that does not conform to JCB guidelines will delay the acceptance of your manuscript.

- 1) Text limits: Character count for Articles is < 40,000, not including spaces. Count includes abstract, introduction, results, discussion, and acknowledgments. Count does not include title page, figure legends, materials and methods, references, tables, or supplemental legends.
- 2) Figures limits: Articles may have up to 10 main figures and 5 supplemental figures/tables.
- 3) Figure formatting: Scale bars must be present on all microscopy images, including inset magnifications. Molecular weight or nucleic acid size markers must be included on all gel electrophoresis. Please avoid pairing red and green for images and graphs to ensure legibility for color-blind readers. If red and green are paired for images, please ensure that the particular red and green hues used in micrographs are distinctive with any of the colorblind types. If not, please modify colors accordingly or provide separate images of the individual channels.
- 4) Statistical analysis: Error bars on graphic representations of numerical data must be clearly described in the figure legend. The number of independent data points (n) represented in a graph must be indicated in the legend. Statistical methods should be explained in full in the materials and methods. For figures presenting pooled data the statistical measure should be defined in the figure legends. Please also be sure to indicate the statistical tests used in each of your experiments (either in the figure legend itself or in a separate methods section) as well as the parameters of the test (for example, if you ran a t-test, please indicate if it was one- or two-sided, etc.). Also, if you used parametric tests, please indicate if the data distribution was tested for normality (and if so, how). If not, you must state something to the effect that "Data distribution was assumed to be normal but this was not formally tested."
- 5) Abstract and title: The abstract should be no longer than 160 words and should communicate the significance of the paper for a general audience. The title should be less than 100 characters including spaces. Make the title concise but accessible to a general readership.
** For clarity we suggest changing the title to:
"I κ B kinase phosphorylates cytoplasmic 1 TDP-43 and promotes its proteasomal degradation"
- 6) Materials and methods: Should be comprehensive and not simply reference a previous publication for details on how an experiment was performed. Please provide full descriptions in the text for readers who may not have access to referenced manuscripts. We also provide a report from SciScore and an associate score, which we encourage you to use as a means of evaluating and improving the methods section.
** Please provide brief details on wt and 2A3S plasmids, and on AAV preparation.
- 7) Please be sure to provide the sequences for all of your primers/oligos and RNAi constructs in the materials and methods. You must also indicate in the methods the source, species, and catalog numbers (where appropriate) for all of your antibodies.

Please also indicate the acquisition and quantification methods for immunoblotting/western blots.

8) Microscope image acquisition: The following information must be provided about the acquisition and processing of images:

- a. Make and model of microscope
- b. Type, magnification, and numerical aperture of the objective lenses
- c. Temperature
- d. Imaging medium
- e. Fluorochromes
- f. Camera make and model
- g. Acquisition software
- h. Any software used for image processing subsequent to data acquisition. Please include details and types of operations involved (e.g., type of deconvolution, 3D reconstitutions, surface or volume rendering, gamma adjustments, etc.).

10) Supplemental materials: There are strict limits on the allowable amount of supplemental data. Articles may have up to 5 supplemental figures. Please also note that tables, like figures, should be provided as individual, editable files. A summary of all supplemental material should appear at the end of the Materials and methods section.

13) ORCID IDs: ORCID IDs are unique identifiers allowing researchers to create a record of their various scholarly contributions in a single place. At resubmission of your final files, please include an ORCID ID for all authors.

Please note that JCB now requires authors to submit Source Data used to generate figures containing gels and Western blots with all revised manuscripts. This Source Data consists of fully uncropped and unprocessed images for each gel/blot displayed in the main and supplemental figures. Since your paper includes cropped gel and/or blot images, please be sure to provide one Source Data file for each figure that contains gels and/or blots along with your revised manuscript files. File names for Source Data figures should be alphanumeric without any spaces or special characters (i.e., SourceDataF#, where F# refers to the associated main figure number or SourceDataFS# for those associated with Supplementary figures). The lanes of the gels/blots should be labeled as they are in the associated figure, the place where cropping was applied should be marked (with a box), and molecular weight/size standards should be labeled wherever possible. Source Data files will be directly linked to specific figures in the published article.

Journal of Cell Biology now requires a data availability statement for all research article submissions. These statements will be published in the article directly above the Acknowledgments. The statement should address all data underlying the research presented in the manuscript. Please visit the JCB instructions for authors for guidelines and examples of statements at (<https://rupress.org/jcb/pages/editorial-policies#data-availability-statement>).

WHEN APPROPRIATE: The source code for all custom computational methods published in JCB must be made freely available as supplemental material hosted at www.jcb.org. Please contact the JCB Editorial Office to find out how to submit your custom macros, code for custom algorithms, etc. Generally, these are provided as raw code in a .txt file or as other file types in a .zip file. Please also include a one-sentence summary of each file in the Online Supplemental Material paragraph of your manuscript.

B. FINAL FILES:

Thank you for this interesting contribution, we look forward to publishing your paper in Journal of Cell Biology.

Sincerely,

Zu-Hang Sheng
Monitoring Editor
Journal of Cell Biology

Tim Fessenden
Scientific Editor
Journal of Cell Biology

Reviewer #1 (Comments to the Authors (Required)):

I see the authors have made tremendous efforts and done many additional experiments to address the questions that I (and the other reviewers) raised. The manuscript has been substantially improved, especially with the new data of the experiments on the endogenous TDP-43, the in vitro phosphorylation assay and the cytotoxicity tests. Thus, I'd like to recommend this paper for publication in the Journal of Cell Biology.

Reviewer #3 (Comments to the Authors (Required)):

The revised manuscript has addressed most of my concerns. However, there are still a few concerns that need to be addressed before the manuscript can be accepted.

1) As the authors point out, since the WT endogenous TDP-43 is localized to the nucleus, it's hard to detect the effect of IKKbeta on TDP-43. It will be important to use mutant TDP-43 with NLS mutation or deletion to determine whether IKKbeta has any effect on cytoplasmic localized WT TDP-43 and compare its effect on aggregation prone TDP-43.

2) Fig. 2E-G MG132 has a similar effect on TDP-43 when IKK beta WT or SA is expressed, which means even the TDP-43 is not phosphorylated by Ikk-beta, it's still subject to proteasome mediated degradation?

- 3) Fig. 5E and 5F: it will be nice to merge the two graphs into one graph to compare WT and S92D mutant.
- 4) Fig. 8: What does the asterisk indicate? It should be explained in the figure legend. If it's lipofuscin, how do the authors differentiate lipofuscin vs real immunostaining signals? The sections should be treated with TRUE BLACK or other chemicals to get rid of lipofuscin during the staining procedure.
- 5) Fig. 10 The number of mice used in the study is very small considering the variability typically seen with AAV injection and mediated expression. At least the relative expression of TDP-43 and IKK should be quantified to make sure that proteins are expressed at similar levels.